# p62 sorts Lupus La and selected microRNAs into breast cancer-derived exosomes

Jordan Matthew Ngo[1] , Justin Krish Williams[1] , Morayma Mercedes Temoche-Diaz[2] , Abinayaa Murugupandiyan[1] , and Randy Schekman[1,3]

**Exosomes are multivesicular body-derived extracellular vesicles that are secreted by metazoan cells. Exosomes have utility as disease biomarkers, and exosome-mediated miRNA secretion has been proposed to facilitate tumor growth and metastasis. Previously, we demonstrated that the Lupus La protein (La) mediates the selective incorporation of miR-122 into metastatic breast cancer–derived exosomes; however, the mechanism by which La itself is sorted into exosomes remains unknown. Using unbiased proximity labeling proteomics, biochemical fractionation, superresolution microscopy, and genetic tools, we establish that the selective autophagy receptor p62 sorts La and miR-122 into exosomes. We then performed small RNA sequencing and found that p62 depletion reduces the exosomal secretion of tumor suppressor miRNAs and results in their accumulation within cells. Our data indicate that p62 is a quality control factor that modulates the miRNA composition of exosomes. Cancer cells may exploit p62-dependent exosome cargo sorting to eliminate tumor suppressor miRNAs and thus to promote cell proliferation.**

## Introduction

Extracellular vesicles (EVs) are membrane-bounded structures that are exported to the extracellular milieu of cells in culture and *in vivo*. Eukaryotic cells secrete a diverse array of EVs, which can be broadly classified into two distinct subpopulations on the basis of their membrane of origin: microvesicles and exosomes (Colombo et al., 2014; Van Niel et al., 2018). Microvesicles are EVs that bud outward from the plasma membrane, such as apoptotic bodies and protrusion-derived vesicles (Cocucci et al., 2009; Tricarico et al., 2016). Exosomes instead originate from the endocytic pathway. Inward budding of the late endosome limiting membrane produces intraluminal vesicles (ILVs), which are secreted as exosomes upon multivesicular body (MVB) exocytosis (Harding et al., 1983; Pan and Johnstone, 1983).

EVs have elicited significant interest in the field of cancer biology, as cancer cells produce more EVs than non-transformed cells (Szczepanski et al., 2011; Rodríguez et al., 2014). EVs have been suggested to participate in diverse cancer processes such as tumor growth, immune evasion, and metastasis (Hoshino et al., 2015; Hong et al., 2016; Kalluri, 2016; Kosaka et al., 2016; Lobb et al., 2017). Multiple studies have also proposed that exosome-mediated miRNA transfer facilitates metastasis by preparing a pre-metastatic niche before tumor cell arrival (Zhou et al., 2014; Fong et al., 2015; Hoshino et al., 2015; Tominaga et al., 2015;

Peinado et al., 2017; Cao et al., 2022). As a result, the molecular mechanisms that mediate the capture of RNA molecules into EVs have received significant attention.

EV sequencing studies have demonstrated that the RNA profile of EVs, particularly exosomes, is distinct from that of their progenitor cells (Tosar et al., 2015; Shurtleff et al., 2017; Upton et al., 2021). Certain RNA transcripts, such as miRNAs, tRNAs, and yRNAs, are enriched within purified exosome preparations, suggesting the presence of high-fidelity pathways that sort RNA molecules into exosomes. Correspondingly, multiple RNA-binding proteins (RBPs) have been suggested to modulate the RNA composition of exosomes (Villarroya-Beltri et al., 2013; Mukherjee et al., 2016; Santangelo et al., 2016; Teng et al., 2017). However, some of these studies isolated exosomes using high-speed sedimentation procedures that also collected microvesicles and non-vesicular ribonucleoprotein (RNP) complexes. To address this, we have developed buoyant density gradient–based fractionation procedures to separate exosomes, microvesicles, and non-vesicular RNPs (Shurtleff et al., 2016; Temoche-Diaz et al., 2019; Liu et al., 2021). Using these methods, we found that the Lupus La protein (La) mediates the selective incorporation of miR-122 and other miRNAs into high buoyant density, CD63-positive exosomes isolated from the highly invasive breast cancer cell line MDA-MB-231 (Temoche-

---

[1]Department of Molecular and Cell Biology, University of California, Berkeley, Berkeley, CA, USA;   [2]Department of Plant and Microbial Biology, University of California, Berkeley, Berkeley, CA, USA;   [3]Howard Hughes Medical Institute, University of California, Berkeley, Berkeley, CA, USA.

Correspondence to Randy Schekman: schekman@berkeley.edu

M.M. Temoche-Diaz's current affiliation is TornadoBio, San Francisco, CA, USA.

Diaz et al., 2019). However, the molecular mechanism by which La itself is incorporated into exosomes remains unknown.

In this study, we sought to elucidate the molecular mechanism by which La is sorted into breast cancer–derived exosomes. Our studies revealed that LC3 lipidation permits the recruitment of the selective autophagy receptor p62 to late endosomes, where it functions to sequester La, miR-122 and a subset of other miRNAs into ILVs. We also found that many miRNAs that require p62 for their exosomal secretion and accumulate in p62-deficient cells have been implicated in the suppression of tumor cell growth. These data contribute to ongoing discussion in the EV field concerning whether EVs serve to deliver luminal cargo to the cytoplasm of recipient cells or to eliminate unwanted cellular material that may compromise cellular function or homeostasis. Our identification of p62 as a factor that selectively sequesters tumor suppressor miRNAs into breast cancer–derived exosomes provides support for a role of EVs in the elimination of unwanted cellular material.

## Results

### Cytoplasmic La is captured into the lumen of late endosomes

To elucidate the mechanism by which La is secreted in exosomes, we first sought to evaluate the subcellular distribution of La. Previous studies have established that La primarily resides within the nucleus at steady state (Hendrick et al., 1981; Wolin and Cedervall, 2002). However, La has been documented to traffic to and exert function in the cytoplasm (Cardinali et al., 2003; Intine et al., 2003; Petz et al., 2012). To assess the distribution of cytoplasmic La, we mechanically ruptured MDA-MB-231 cells and employed differential centrifugation (Fig. 1, A and B). We observed that La co-fractionated with membrane pellets containing CD63, Rab5, and LAMP1. We then conducted proteinase K protection experiments using a 20,000 × g sedimentable membrane fraction to elucidate whether a portion of cytoplasmic La was sequestered within a membrane-protected compartment (Fig. 1, C and D). We observed that ∼4% of cytoplasmic La was resistant to proteinase K digestion. Titration of proteinase K (up to 40 μg/ml) did not diminish the amount of membrane-enclosed La. Tim23, a mitochondrial inner membrane protein, and Rab5, a membrane-anchored Rab GTPase that is exposed to the cytoplasm, served as negative and positive controls for proteinase K accessibility, respectively. Solubilization of the 20,000 × g membrane fraction with the nonionic detergent Triton X-100 (TX-100) rendered all proteins tested sensitive to proteinase K-mediated digestion at the lowest tested concentration (20 μg/ml). These results demonstrated that a portion of cytoplasmic La is present within the lumen of a membrane-bound compartment.

We next sought to determine the subcellular localization of endogenous, cytoplasmic La using superresolution microscopy. Given that a majority of La resides within the nucleus at steady state, we sought to establish a cell permeabilization strategy that would allow us to visually distinguish between the nuclear and cytoplasmic pools of La. Saponin is a plant-derived glycoside that selectively permeabilizes cholesterol-rich membranes, such as the plasma membrane, but not the nuclear membrane (Jamur

and Oliver, 2010). We thus reasoned that the use of saponin-permeabilized cells would permit visual analysis of cytoplasmic, rather than nuclear, La. To test this hypothesis, we permeabilized cells with TX-100 (0.1%) or saponin (0.02%) and visualized endogenous La (Fig. 1 E and Fig. S1 A). La was detected primarily within the nucleus of TX-100-permeabilized cells. In contrast, La was detected only in the cytoplasm of saponin-permeabilized cells. These results indicate that saponin-permeabilized cells permit visual discrimination between nuclear and cytoplasmic La.

We then investigated whether cytoplasmic La could be captured into the lumen of endogenous CD63-positive late endosomes. Using our saponin-based cell permeabilization protocol, we visualized endogenous La and CD63 and observed the presence of La puncta within the lumen of CD63-positive late endosomes (Fig. 1, F and G; and Fig. S1 B). We also investigated whether cytoplasmic La co-localized with the P-body marker, DDX6, as we previously demonstrated that liquid–liquid phase separation contributes to the sorting of miRNAs into exosomes (Liu et al., 2021). We visualized endogenous La and DDX6 and observed that cytoplasmic La partially co-localized with DDX6 (Fig. S2 A). Altogether, our membrane fractionation and superresolution microscopy data demonstrate that cytoplasmic La can be captured into the lumen of late endosomes.

### Identification of candidate La sorting factors by unbiased proximity labeling

After determining that La can be captured into the lumen of late endosomes, we sought to identify candidate proteins that facilitate this process. Proximity labeling using promiscuous enzymes such as TurboID and APEX2 permits the unbiased identification of protein interaction networks and organelle proteomes (Qin et al., 2021). Upon the addition of biotin phenol and a short pulse of hydrogen peroxide ($H_2O_2$), the evolved soybean ascorbate peroxidase APEX2 generates short-lived, membrane-impermeable biotin-phenoxyl radicals that form covalent adducts with substrates within a 10–20 nm radius (Lam et al., 2015). These biotinylated substrates can then be enriched using streptavidin-conjugated beads and analyzed by mass spectrometry. Given the demonstrated utility of APEX2 in defining organellar proteomes (Rhee et al., 2013; Hung et al., 2014; Hung et al., 2017; Bersuker et al., 2018), we sought to employ APEX2 to identify factors that contribute to the capture of La into exosomes. We generated MDA-MB-231 cell lines that express APEX2 or APEX2 fused to La (La-APEX2), conducted APEX2 labeling, fractionated cells by sedimentation to obtain membranes, and purified the biotinylated proteins using streptavidin-conjugated magnetic beads for mass spectrometry analysis (Fig. 2 A and Table S1).

Gene ontology (GO) analysis for the subcellular localization of proteins enriched in the La-APEX2 sample over APEX2 revealed GO terms related to membrane trafficking and exosome biogenesis, such as "late endosome" and "extracellular exosome" (Fig. 2 B). We then evaluated our mass spectrometry results to identify proteins that have been reported to recognize and/or sort cargo in various biological processes. Among our mass spectrometry results, we identified multiple RBPs, such as

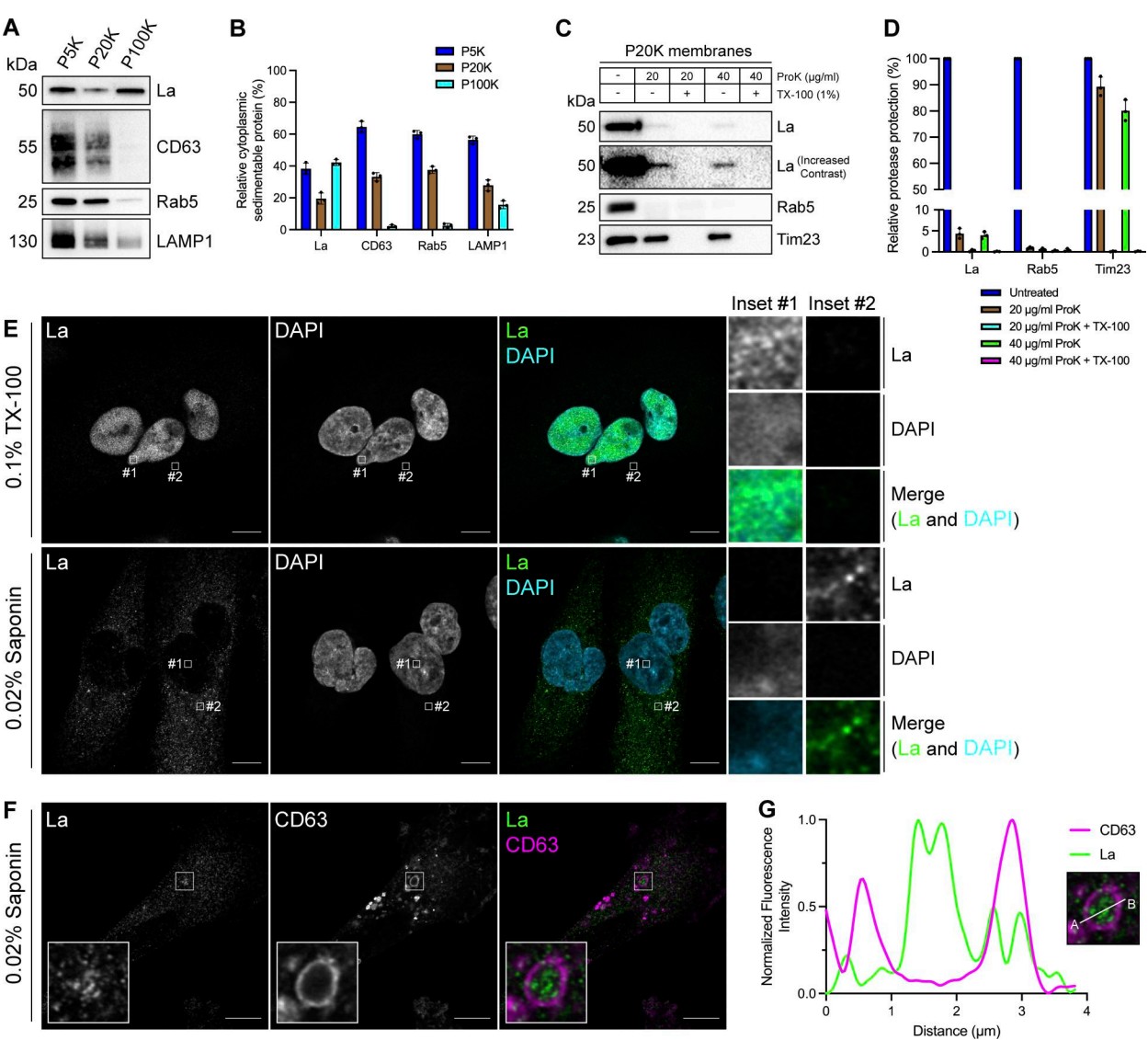

Figure 1. **Cytosolic La is captured into the lumen of late endosomes. (A)** Mechanically ruptured MDA-MB-231 cells were subjected to differential centrifugation. Immunoblot analysis of the 5,000 × g pellet (P5K), 20,000 × g pellet (P20K), and 100,000 × g pellet (P100K) was conducted to assess the presence of the indicated proteins. **(B)** Quantification of the indicated proteins within the membrane pellet fractions from Fig. 1 A (n = 3). **(C)** Immunoblot analysis of proteinase K protection assays conducted on a P20K membrane fraction to assess whether the indicated proteins were sequestered within the lumen of a detergent-sensitive compartment. **(D)** Quantification of the proteinase K protection experiments from Fig. 1 C (n = 3). **(E)** Airyscan microscopy of endogenous La with DAPI counterstain from MDA-MB-231 WT cells permeabilized with either 0.1% TX-100 or 0.02% saponin. Green: La; cyan: DAPI. Scale bar: 10 μm. **(F)** Airyscan microscopy of endogenous La and CD63 from MDA-MB-231 WT cells permeabilized with 0.02% saponin. Green: La; magenta: CD63. Scale bar: 10 μm. **(G)** Quantification of La and CD63 fluorescence intensity from point A to point B in the indicated inset of Fig. 1F. Source data are available for this figure: SourceData F1.

hnRNPA2B1 and HuR/ELAVL1, that have been reported to sort miRNAs into EVs (Villarroya-Beltri et al., 2013; Mukherjee et al., 2016). Of particular interest, we identified the selective autophagy receptor, p62 (also known as SQSTM1), as a La-APEX2 unique candidate (Fig. 2 C). p62 was the first identified selective autophagy receptor and interacts with LC3 by virtue of its LC3-interaction region (LIR) motif (Bjørkøy et al., 2005; Pankiv et al., 2007; Lamark and Johansen, 2021). Previously, Leidal et al. discovered that a subset of lipidated LC3 localizes to late endosomes and mediates the selective capture of RBPs into EVs (Leidal et al., 2020). We thus hypothesized that p62 sorts La into exosomes.

## p62 is captured together with La into the lumen of late endosomes

After identifying p62 as a candidate La sorting factor, we sought to assess the subcellular distribution of endogenous p62. We performed differential centrifugation on mechanically ruptured cells to evaluate the distribution of p62 relative to various endolysosomal markers (Fig. 3, A and B). We observed that p62 co-fractionated with membrane pellets containing CD63, Rab5, and LAMP2A. Proteinase K protection experiments using a 20,000 × g membrane fraction revealed that ~25% of cytoplasmic p62 was resistant to proteinase K digestion in the absence of TX-100 (Fig. 3, C and D). We then inquired whether p62 was captured

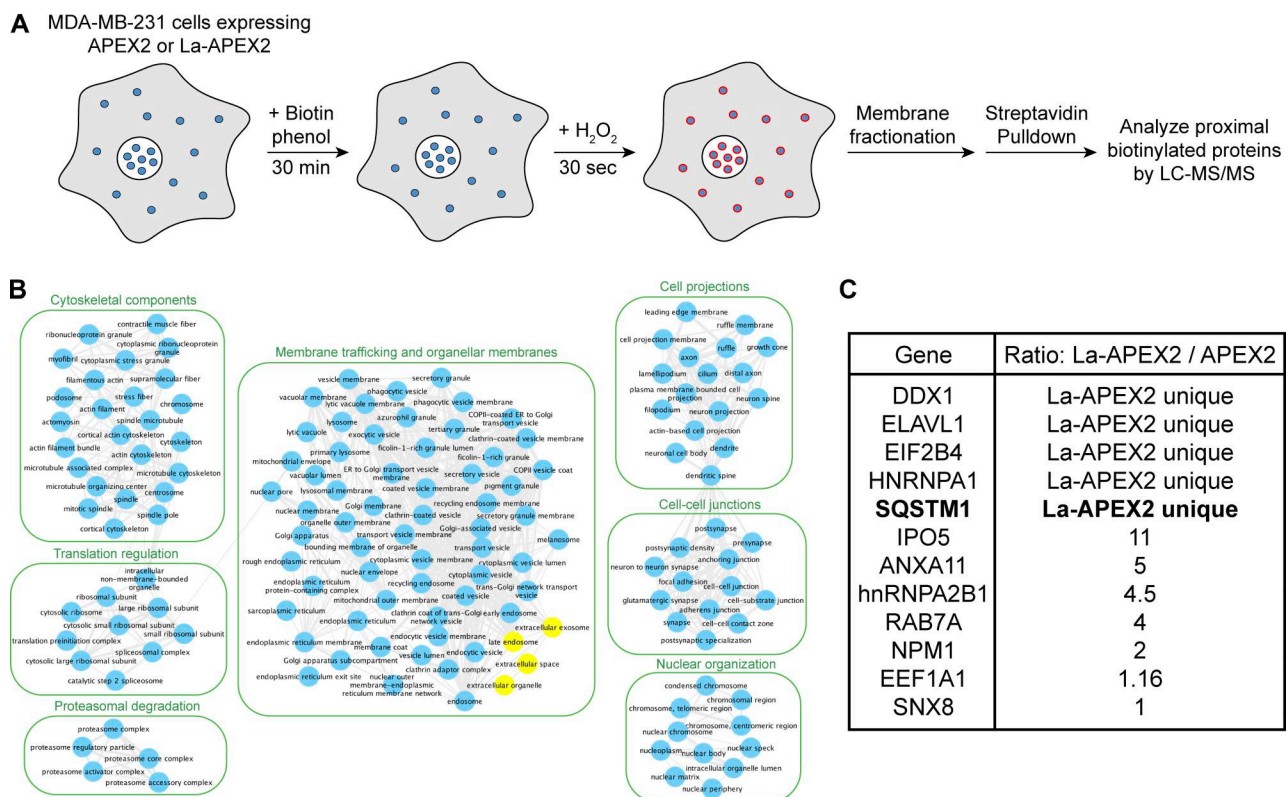

Figure 2. **Identification of candidate La sorting factors by unbiased proximity labeling. (A)** Schematic of the proximity labeling strategy to identify La sorting factors. APEX2 or La-APEX2 proximal proteins were labeled with biotin upon addition of biotin phenol and $H_2O_2$. The cells were then fractionated to obtain a 20,000 × *g* membrane pellet, and the biotinylated proteins were purified and identified by mass spectrometry. **(B)** GO analysis for the subcellular localization of proteins that were enriched more than twofold in La-APEX2 compared with APEX2. GO terms for late endosome, extracellular exosome, "extracellular organelle," and "extracellular space" are colored yellow. **(C)** Table listing proteins identified in the APEX2 proximity labeling experiment. The ratio of peptides between La-APEX2 and APEX2 for each hit is listed.

together with La into the lumen of CD63-positive late endosomes. We visualized the subcellular localization of endogenous p62, La, and CD63 in saponin-permeabilized cells using super-resolution microscopy. p62 was primarily diffuse but also observed in cytoplasmic puncta. A small portion of these p62 puncta co-localized with La inside the lumen of CD63-positive late endosomes (Fig. 3, E and F; and Fig. S1 C). We then asked whether p62 could be captured together with LC3 into the lumen of CD63-positive late endosomes. We visualized endogenous p62, LC3, and CD63 in saponin-permeabilized cells and observed p62/LC3 double-positive puncta at both the limiting membrane and within the lumen of CD63-positive late endosomes (Fig. 3, G and H; and Fig. S1 D). These data suggest that a small portion of p62 is captured together with La into the lumen of CD63-positive late endosomes.

## p62 is secreted within the lumen of exosomes

After confirming that p62 can be captured together with La into the lumen of late endosomes, we sought to elucidate whether p62 was secreted in exosomes. We previously developed an approach to immunoprecipitate (IP) CD63-positive exosomes from HEK293T cells (Shurtleff et al., 2016). However, this CD63 antibody was unable to IP exosomes derived from MDA-MB-231 cells. We therefore sought to establish a universal exosome

affinity purification strategy. Verweij et al. previously inserted pHluorin, a pH-sensitive GFP variant, into the first extracellular loop (ECL1) of CD63 to visualize the fusion of acidic MVBs with the plasma membrane using total internal reflection fluorescence microscopy (Verweij et al., 2018). The authors reported that CD63-pHluorin[ECL1] expression did not affect CD63 trafficking, localization or MVB morphology. We thus employed a similar strategy and inserted a HA epitope tag at the N terminus of CD63 and monomeric enhanced GFP (mEGFP) between Gln36 and Leu37 (HA-CD63-mEGFP[ECL1]) (Fig. 4 A). If correctly inserted into the late endosome limiting membrane, mEGFP should be poised toward the organelle lumen. Conversely, mEGFP should be exposed at the exosome surface due to the topology inversion that occurs during ILV biogenesis. We reasoned that this strategy should allow us to immunoisolate CD63-positive exosomes using magnetic beads conjugated to the anti-GFP nanobody. This strategy exploits the tight binding affinity of the anti-GFP nanobody for GFP (dissociation constant [$K_d$] of 1 pM) to permit efficient isolation of HA-CD63-mEGFP[ECL1] exosomes.

We generated an MDA-MB-231 cell line that stably expressed HA-CD63-mEGFP[ECL1] (Fig. 4 B). HA-CD63-mEGFP[ECL1] migrated as a single band at an apparent molecular weight of 55 kDa, consistent with the previously reported CD63-pHluorin[ECL1] (Verweij et al., 2018). We then employed two complementary

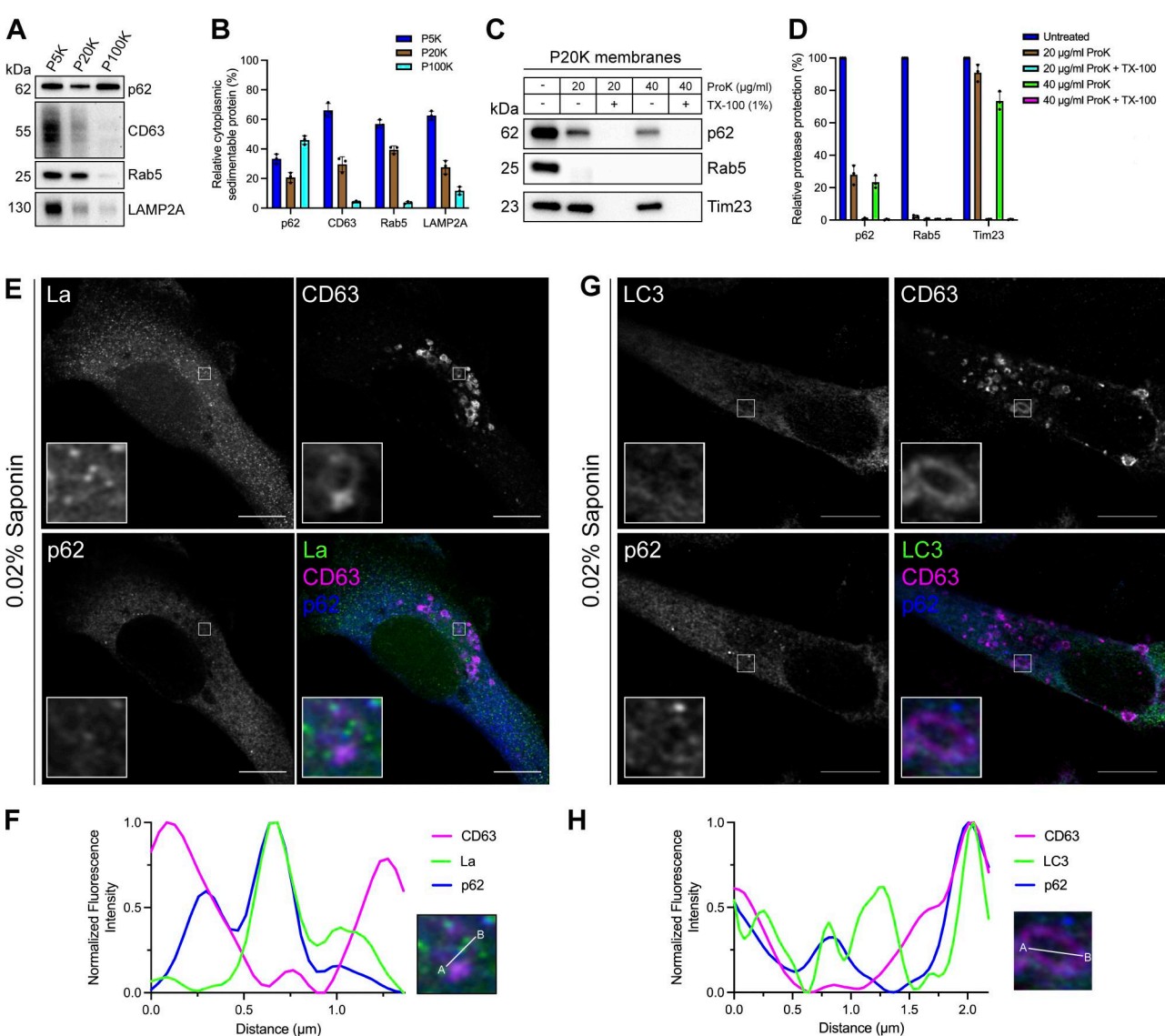

Figure 3. **p62 is captured together with La into the lumen of late endosomes**. **(A)** Mechanically ruptured MDA-MB-231 cells were subjected to differential centrifugation. Immunoblot analysis of the 5,000 × *g* pellet (P5K), 20,000 × *g* pellet (P20K), and 100,000 × *g* pellet (P100K) was conducted to evaluate the presence of the indicated proteins. **(B)** Quantification of the indicated proteins within the membrane pellet fractions from Fig. 3 A (*n* = 3). **(C)** Immunoblot analysis of proteinase K protection assays on a P20K membrane fraction to evaluate whether the indicated proteins were sequestered within the lumen of a detergent-sensitive compartment. **(D)** Quantification of the proteinase K protection experiments from Fig. 3 C (*n* = 3). **(E)** Airyscan microscopy of endogenous La, CD63, and p62 from MDA-MB-231 cells permeabilized with 0.02% saponin. Green: La; magenta: CD63; blue: p62. Scale bar: 10 µm. **(F)** Quantification of La, CD63, and p62 fluorescence intensities from point A to point B of the indicated inset of Fig. 3 E. **(G)** Airyscan microscopy of endogenous LC3, CD63, and p62 from MDA-MB-231 cells permeabilized with 0.02% saponin. Green: LC3; magenta: CD63; blue: p62. Scale bar: 10 µm. **(H)** Quantification of LC3, CD63, and p62 fluorescence intensities from point A to point B of the indicated inset of Fig. 3 G. Source data are available for this figure: SourceData F3.

approaches to validate whether HA-CD63-mEGFP$^{ECL1}$ was correctly sorted and topologically oriented within the late endosome limiting membrane. First, we treated HA-CD63-mEGFP$^{ECL1}$ cells with the potent vacuolar ATPase (V-ATPase) inhibitor, bafilomycin A1 (BafA1). Upon binding to the central V-ATPase subunit ATP6V0C, BafA1 inhibits proton translocation and subsequently deacidifies cells (Wang et al., 2021). We reasoned that BafA1 treatment should dequench HA-CD63-mEGFP$^{ECL1}$ fluorescence by neutralizing the acidic pH of the late endosome lumen (Miesenböck et al., 1998). As expected, we observed that BafA1 treatment deacidified late endosomes and led to a

concomitant increase in HA-CD63-mEGFP$^{ECL1}$ fluorescence (Fig. 4 C). This result indicated that HA-CD63-mEGFP$^{ECL1}$ expression did not impair endolysosomal acidification. We next conducted an organelle IP using magnetic beads conjugated to either an anti-HA antibody or the anti-GFP nanobody. We reasoned that anti-HA beads should enrich more late endosomes than anti-GFP beads. Indeed, we observed that anti-HA beads enriched more LAMP2A-positive late endosomes compared with anti-GFP beads (Fig. 4 D). These orthogonal approaches provide evidence that HA-CD63-mEGFP$^{ECL1}$ is properly sorted and oriented within the late endosome limiting membrane.

Figure 4. **p62 is secreted within the lumen of immunoisolated exosomes**. **(A)** Domain architecture and schematic illustrating the membrane topology of HA-CD63-mEGFP[ECL1] when inserted into the limiting membrane of endosomes (left) and exosomes (right). mEGFP should be quenched within the acidic lumen of late endosomes. **(B)** Immunoblot analysis of MDA-MB-231 cells expressing HA-CD63-mEGFP[ECL1]. **(C)** Fluorescence microscopy of MDA-MB-231 cells expressing HA-CD63-mEGFP[ECL1] upon treatment with either DMSO or BafA1 (100 nM) for 16 h. Green: HA-CD63-mEGFP[ECL1]; magenta: LysoTracker. **(D)** Immunoblot analysis of a PNS input and late endosome IPs from lysed HA-CD63-mEGFP[ECL1] cells. **(E)** Schematic illustrating the procedure to immunoisolate exosomes from HA-CD63-mEGFP[ECL1] cells. **(F)** Immunoblot analysis of a HSP input and exosome IPs from the conditioned medium of HA-CD63-mEGFP[ECL1] cells. **(G)** Immunoblot analysis of an extracellular HSP input, flow-through, and anti-GFP IPs from the conditioned medium of HA-CD63-mEGFP[ECL1] cells. **(H)** Immunoblot analysis of proteinase K protection assays performed on immunoisolated exosomes to assess whether the indicated proteins are sequestered within the lumen of exosomes. TAMRA-labeled α-synuclein was added as an extravesicular spike to confirm complete proteolysis. **(I)** Quantification of the proteinase K protection experiments from Fig. 4 H ($n$ = 3). Source data are available for this figure: SourceData F4.

After validating the topology of HA-CD63-mEGFP[ECL1] in late endosomes, we evaluated whether HA-CD63-mEGFP[ECL1] trafficked to and assumed the correct topology in exosomes. To accomplish this, we employed a strategy consisting of differential centrifugation followed by IP (Fig. 4 E). We obtained conditioned medium from HA-CD63-mEGFP[ECL1] cultured cells, utilized low- and medium-speed centrifugation to remove cells and large debris, then employed high-speed sedimentation to obtain an extracellular high-speed pellet (HSP) fraction. We then conducted reciprocal anti-HA and anti-GFP IPs using the HSP fraction to immunopurify exosomes. We observed significant endogenous CD63 using anti-GFP beads but not anti-HA beads (Fig. 4 F). Notably, HA-CD63-mEGFP[ECL1] migrated as a single band, but we immunoisolated exosomes containing endogenous, glycosylated CD63. This provides further evidence that HA-CD63-mEGFP[ECL1] is correctly sorted (together with endogenous CD63) into exosomes and could be used as a universal affinity handle for exosome immunoisolation.

We next asked whether p62 was present within purified exosomes (Fig. 4 G). We purified HA-CD63-mEGFP[ECL1] exosomes using our immunoisolation strategy and observed that they were enriched with CD63 and flotillin-2 and diminished of the microvesicle marker, annexin A2 (Jeppesen et al., 2019; Williams et al., 2025). Importantly, we noted that p62 was enriched in our anti-GFP eluate, indicating that extracellular p62 associated with CD63-positive exosomes.

We then performed a proteinase K protection assay on purified exosomes to confirm whether p62 was secreted as a soluble protein that resided within the lumen of exosomes and was not merely associated with the vesicle surface. Proteinase K protection experiments using immunopurified exosomes revealed that exosomal p62 was primarily resistant to proteinase K–mediated digestion in the absence of detergent (Fig. 4, H and I). This is consistent with our previous observation that extracellular La is resistant to proteolysis in the absence of detergent (Temoche-Diaz et al., 2019). Recombinant α-synuclein was

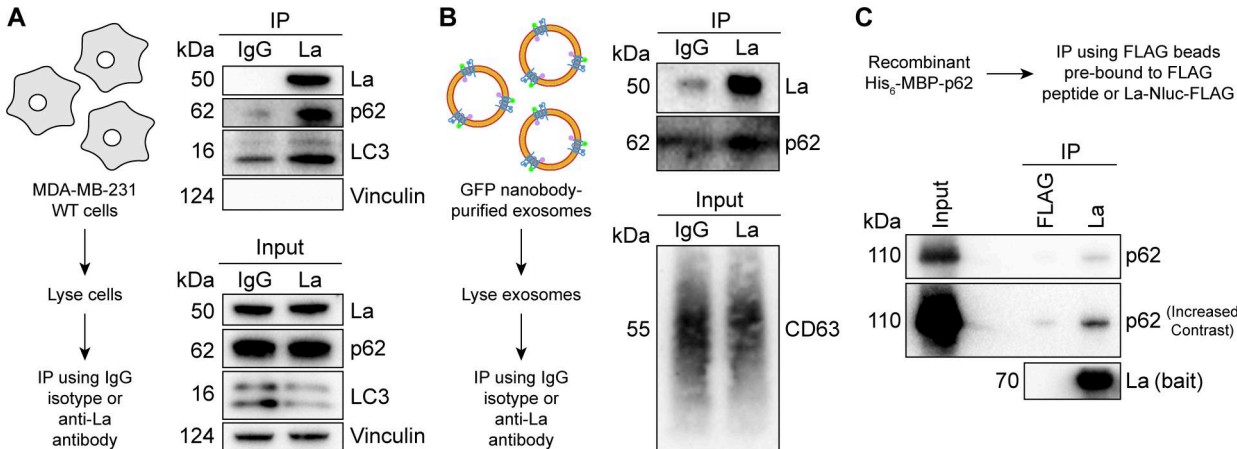

Figure 5. **La interacts with p62 in cells, in immunoisolated exosomes, and _in vitro_. (A)** Immunoblot analysis of PNS inputs and IPs from lysed MDA-MB-231 cells. **(B)** Immunoblot analysis of exosome inputs and IPs from lysed exosomes isolated from the conditioned medium of MDA-MB-231 HA-CD63-mEGFP[ECL1] cells. **(C)** Immunoblot analysis of recombinant protein input and IPs from a La/p62 _in vitro_–binding assay. Source data are available for this figure: SourceData F5.

added as an extravesicular spike to confirm complete proteolysis. Altogether, these data demonstrate that p62 is secreted within the lumen of exosomes.

## La interacts with p62 in cells, in exosomes, and _in vitro_

After confirming that p62 was secreted within the lumen of exosomes, we sought to evaluate the interaction between La and p62. Our proximity labeling experiments suggested that La and p62 were located in close proximity within cells but did not confirm whether they interacted (Fig. 2 C). We therefore utilized IP experiments to elucidate whether La interacted with p62 under native conditions. A postnuclear supernatant (PNS) fraction isolated from MDA-MB-231 WT cells was divided and mixed with either an IgG isotype control antibody or an anti-La antibody. Immunoblot analysis of the immunoprecipitated fractions indicated that p62 and LC3 were enriched in the anti-La IP relative to the IgG control (Fig. 5 A). Vinculin served as a negative control and was greatly diminished in both IPs.

We then evaluated whether La interacted with p62 in exosomes. Exosomes were immunoisolated from HA-CD63-mEGFP[ECL1] cells, lysed, distributed into aliquots, and used for IP experiments. Consistent with our previous result, immunoblot analysis of the immunoprecipitated fractions indicated that the anti-La antibody enriched more p62 relative to the IgG control (Fig. 5 B). We next asked whether La interacts with p62 _in vitro_. To test this, we purified La and p62 from bacterial cells and performed IP experiments (Fig. S3, A and B). Immunoblot analysis of the immunoprecipitated fractions indicated La bound to FLAG beads enriched more p62 relative to the control (Fig. 5 C). These results demonstrate that La interacts with p62 in cells, in exosomes, and _in vitro_.

## La motif and RRM1 mediate the high-affinity interaction between La and miR-122

After validating that La interacts with p62, we sought to characterize the interaction between La and miR-122. We previously identified a bipartite motif in miR-122 (5′ UGGA and 3′ UUU) that is required for La binding (Temoche-Diaz et al., 2019).

However, we did not identify the minimal sequence of La required for its interaction with miR-122. To address this, we performed a series of fluorescent electrophoretic mobility shift assays (EMSAs) using recombinant La expressed and purified from bacterial cells (Fig. S3 A). We first conducted EMSAs with full-length La and either miR-122 or miR-190, a miRNA that is not present within exosomes (Fig. 6 A). Purified La was titrated, mixed with 5′ fluorescently labeled miR-122 or miR-190, incubated at 30°C, resolved by native gel electrophoresis, and visualized by in-gel fluorescence. The measured $K_d$ of La for miR-122 and miR-190 was 14.1 nM and 1.15 µM, respectively (Fig. 6 B).

We next sought to identify the features of La required for its interaction with miR-122. Human La contains an N-terminal La motif (LaM), two RNA recognition motifs (RRM1 and RRM2), and a C-terminal intrinsically disordered region (IDR) (Fig. 6 C) (Wolin and Cedervall, 2002; Martino et al., 2015). We purified a series of truncated La proteins encompassing various combinations of the La functional features (Fig. S2 A). Iterative truncation of the La C terminus revealed that removal of RRM1 greatly decreased the affinity of La for miR-122 (Fig. 6, D and E). Conversely, iterative truncation of the La N terminus demonstrated that LaM was required for the interaction between La and miR-122 (Fig. 6 F). Taken together, these data indicate that LaM and RRM1 form the minimal sequence required for the high-affinity interaction between La and miR-122.

Surprisingly, our EMSA data revealed that removal of the C-terminal IDR increased the affinity of La for miR-122 by ~90-fold (Fig. 6, D and E). Given that the IDR does not directly bind miR-122 (Fig. 6 F), these results indicated that the IDR negatively regulated the interaction between La and miR-122 through structural effects on the remainder of the La protein. We performed competition EMSAs to evaluate the role of the IDR on the La and miR-122 interaction, but we were unable to effectively compete off miR-122 (Fig. S4, A and B).

## p62 and ATG7 are required for La secretion

After characterizing the interaction of La with both p62 and miR-122, we inquired whether p62 was required for the

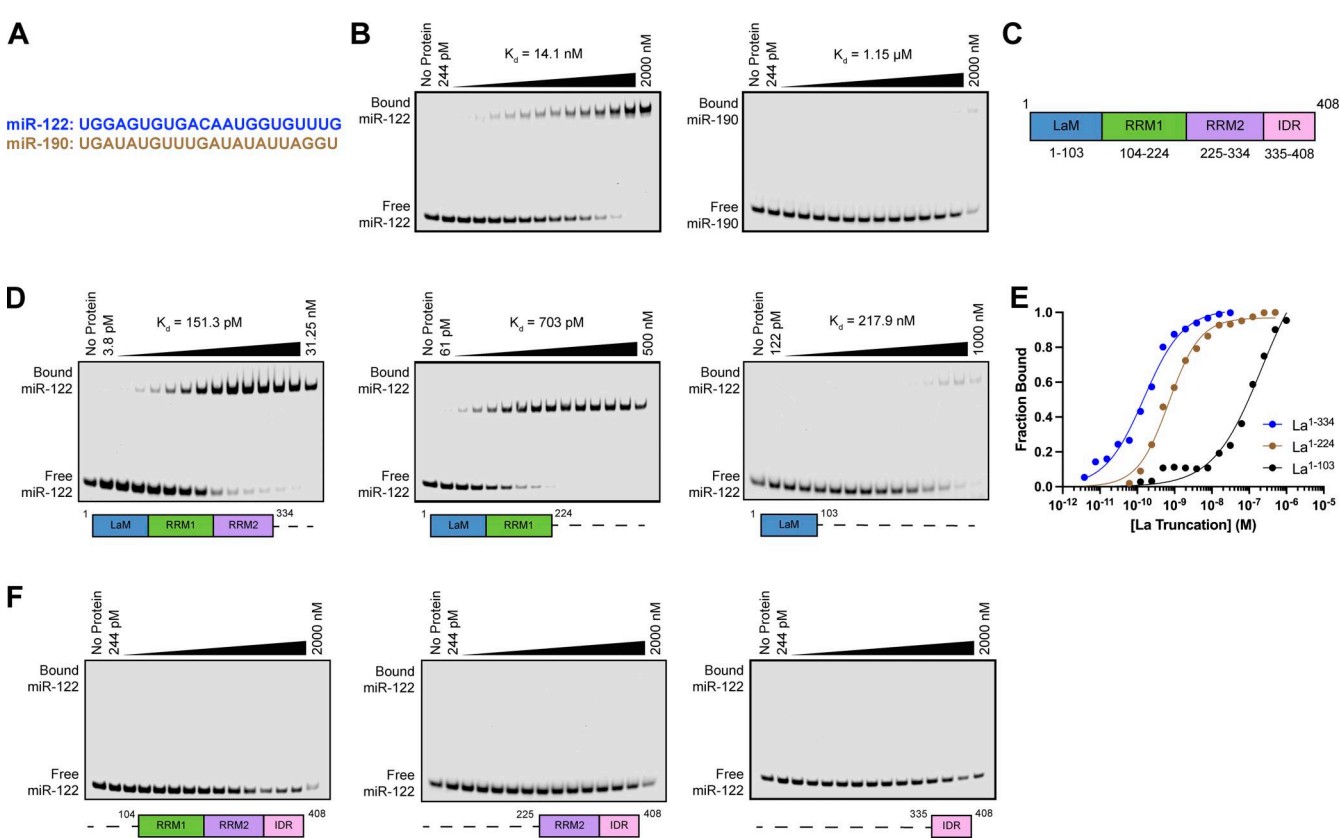

Figure 6. **La interacts directly with miR-122 through LaM and RRM1. (A)** Sequences for miR-122 and miR-190. **(B)** EMSAs with purified La$^{1-408}$ and 5′ fluorescently labeled miR-122 or miR-190. La$^{1-408}$ was titrated from 244 pM to 2 µM. The migration of each miRNA was detected by in-gel fluorescence. The fraction bound was quantified as a function of the exhaustion of free miRNA. **(C)** Schematic illustrating the four functional domains of La. **(D)** EMSAs with 5′ fluorescently labeled miR-122 and purified La$^{1-334}$, La$^{1-224}$, or La$^{1-103}$. The titration range of each La truncation is indicated above each gel. miR-122 migration was detected by in-gel fluorescence. The fraction bound was quantified as a function of the exhaustion of free miR-122. **(E)** Quantification of Fig. 6 D showing the calculated $K_d$ values. **(F)** EMSAs with 5′ fluorescently labeled miR-122 and purified La$^{104-408}$, La$^{225-408}$, or La$^{335-408}$. Each La truncation was titrated from 244 pM to 2 µM. miR-122 migration was detected by in-gel fluorescence. Source data are available for this figure: SourceData F6.

secretion of La. To address this question, we employed CRISPR interference (CRISPRi). CRISPRi attenuates gene expression by repressing gene transcription and does not require the RNA-induced silencing complex (RISC) (Gilbert et al., 2013; Qi et al., 2013). RISC directly binds miRNAs to mediate miRNA-mediated gene repression (Bartel, 2018). We reasoned that the use of CRISPRi instead of RNA interference would prevent miRNA sorting artifacts that could occur due to artificial overload of RISC. We generated HA-CD63-mEGFP$^{ECL1}$ cells that expressed catalytically inactive Cas9 (dCas9) fused to the recently discovered ZIM3 KRAB transcriptional repressor domain (Alerasool et al., 2020). Expression of a single gRNA (sgRNA) targeting SQSTM1 (sgSQSTM1) in these cells effectively depleted p62 relative to a nontargeting sgRNA control (sgNT) (Fig. 7 A).

We next assessed whether p62 depletion affected La secretion. We immunoisolated exosomes from HA-CD63-mEGFP$^{ECL1}$ sgNT and sgSQSTM1 cells and observed that p62 depletion decreased the secretion of exosomal La (Fig. 7, B and C). We then tested whether LC3 lipidation was required for the secretion of La and p62. ATG7 is an autophagy component essential for the ubiquitin-like conjugation of phosphatidylethanolamine to LC3 (Ichimura et al., 2000). We obtained HEK293T ATG7 knockout (KO) cells and verified that ATG7 KO led to a loss of lipidated LC3

(LC3-II) and concomitant accumulation of unlipidated LC3 (LC3-I) (Fig. 7 D). We then assessed whether ATG7 KO attenuated La secretion. We observed that ATG7 KO decreased the secretion of both La and p62 (Fig. 7, E and F). We then asked whether other components of the classical autophagy pathway are required for La and p62 secretion. FIP200 (also known as RB1CC1) is required for autophagosome biogenesis but dispensable for LC3 lipidation (Hara et al., 2008). In contrast to ATG7 KO, FIP200 KO (Fig. 7 G) did not block LC3 lipidation nor attenuate the secretion of La, p62, or LC3 (Fig. 7, H and I). These results indicate that p62 and LC3 lipidation, but not autophagosome formation, are required for La secretion.

**p62 sorts miR-122 and a subset of miRNAs into exosomes**

Finally, we sought to evaluate whether p62 depletion affected the miRNA composition of purified exosomes. We first assessed whether p62 depletion affected the secretion of miR-122 in vesicles sedimented from conditioned medium. RT-qPCR analysis of HSP fractions isolated from HA-CD63-mEGFP$^{ECL1}$ sgNT and sgSQSTM1 cells indicated that p62 depletion reduced miR-122 secretion (Fig. S5 A).

As an unbiased approach to profile exosomal miRNA content, we isolated RNA from MDA-MB-231 HA-CD63-mEGFP$^{ECL1}$ sgNT

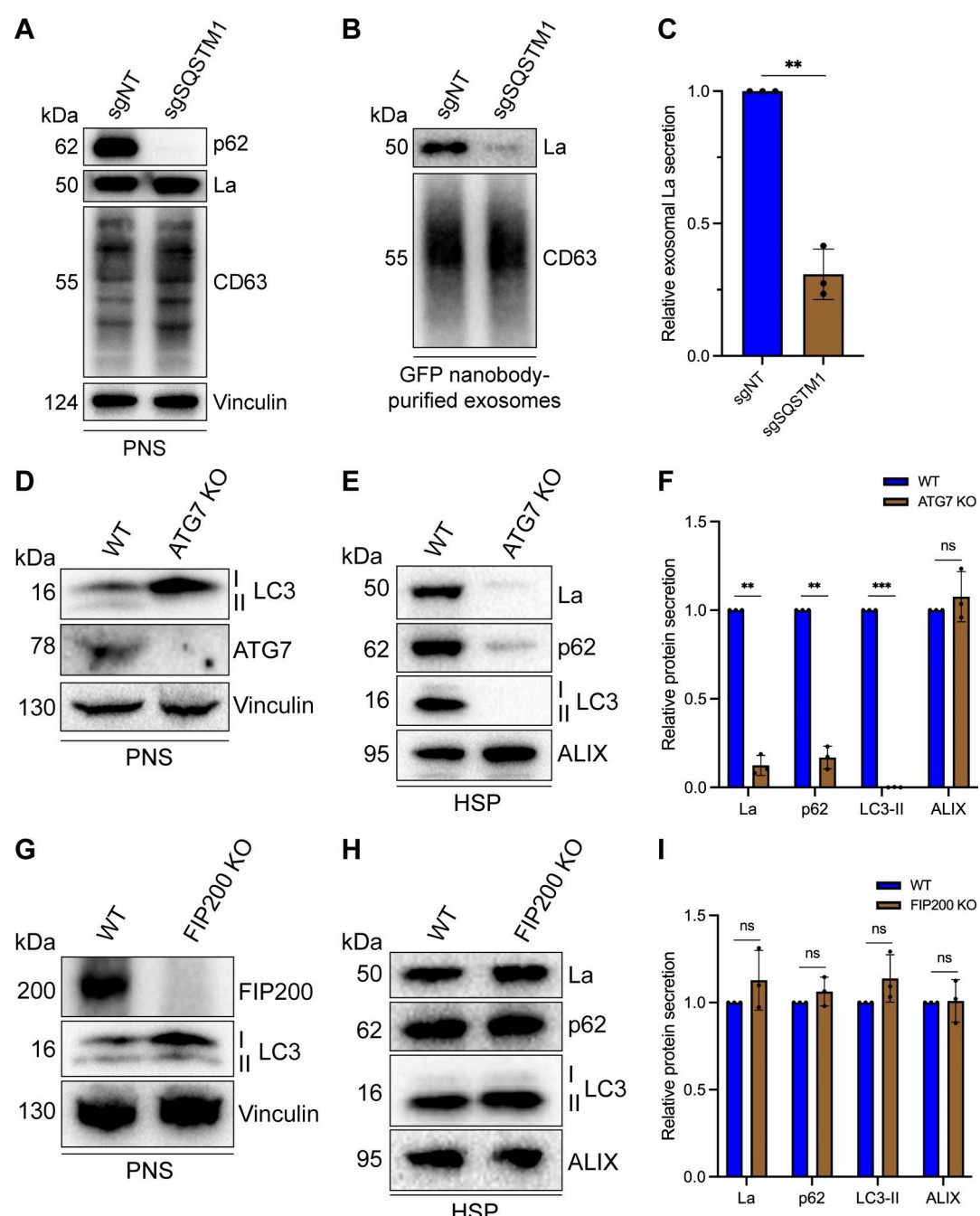

Figure 7. **p62 and ATG7 are required for La secretion. (A)** Immunoblot analysis of MDA-MB-231 cells expressing HA-CD63-mEGFP[ECL1], ZIM3-KRAB-dCas9, and either a nontargeting (sgNT) or p62/SQSTM1-targeting (sgSQSTM1) sgRNA. **(B)** Immunoblot analysis of exosomes immunoisolated from the conditioned medium of MDA-MB-231 cells expressing HA-CD63-mEGFP[ECL1], ZIM3-KRAB-dCas9, and either sgNT or sgSQSTM1. **(C)** Quantification of exosomal La secretion from sgNT and sgSQSTM1 cells. Statistical significance was performed using an unpaired two-tailed $t$ test ($n$ = 3; **P < 0.01). **(D)** Immunoblot analysis of HEK293T WT and ATG7 KO cells. **(E)** Immunoblot analysis of HSP fractions isolated from the conditioned medium of HEK293T WT and ATG7 KO cells. **(F)** Quantification of La, p62, LC3-II, and ALIX secretion from WT and ATG7 KO cells. Statistical significance was performed using unpaired two-tailed $t$ tests ($n$ = 3; **P < 0.01; ***P < 0.001; and ns, not significant). **(G)** Immunoblot analysis of HEK293T WT and FIP200 KO cells. **(H)** Immunoblot analysis of HSP fractions isolated from the conditioned medium of HEK293T WT and FIP200 KO cells. **(I)** Quantification of La, p62, LC3-II, and ALIX secretion from WT and FIP200 KO cells. Statistical significance was performed using unpaired two-tailed $t$ tests ($n$ = 3; ns, not significant). Source data are available for this figure: SourceData F7.

and sgSQSTM1 cells and immunoisolated exosomes, generated cDNA libraries, and performed small RNA sequencing. The sequencing reads were mapped to annotated human miRNAs from miRBase (Kozomara et al., 2019) using the miRDeep2 software (Friedländer et al., 2012), and the mapped

miRNA reads were normalized to the total number of miRNA reads per sample.

We compared the miRNA profile between sgNT and sgSQSTM1 exosomes and observed a decrease in multiple miRNA species (such as miR-33a-5p, miR-122-5p, and miR-191-

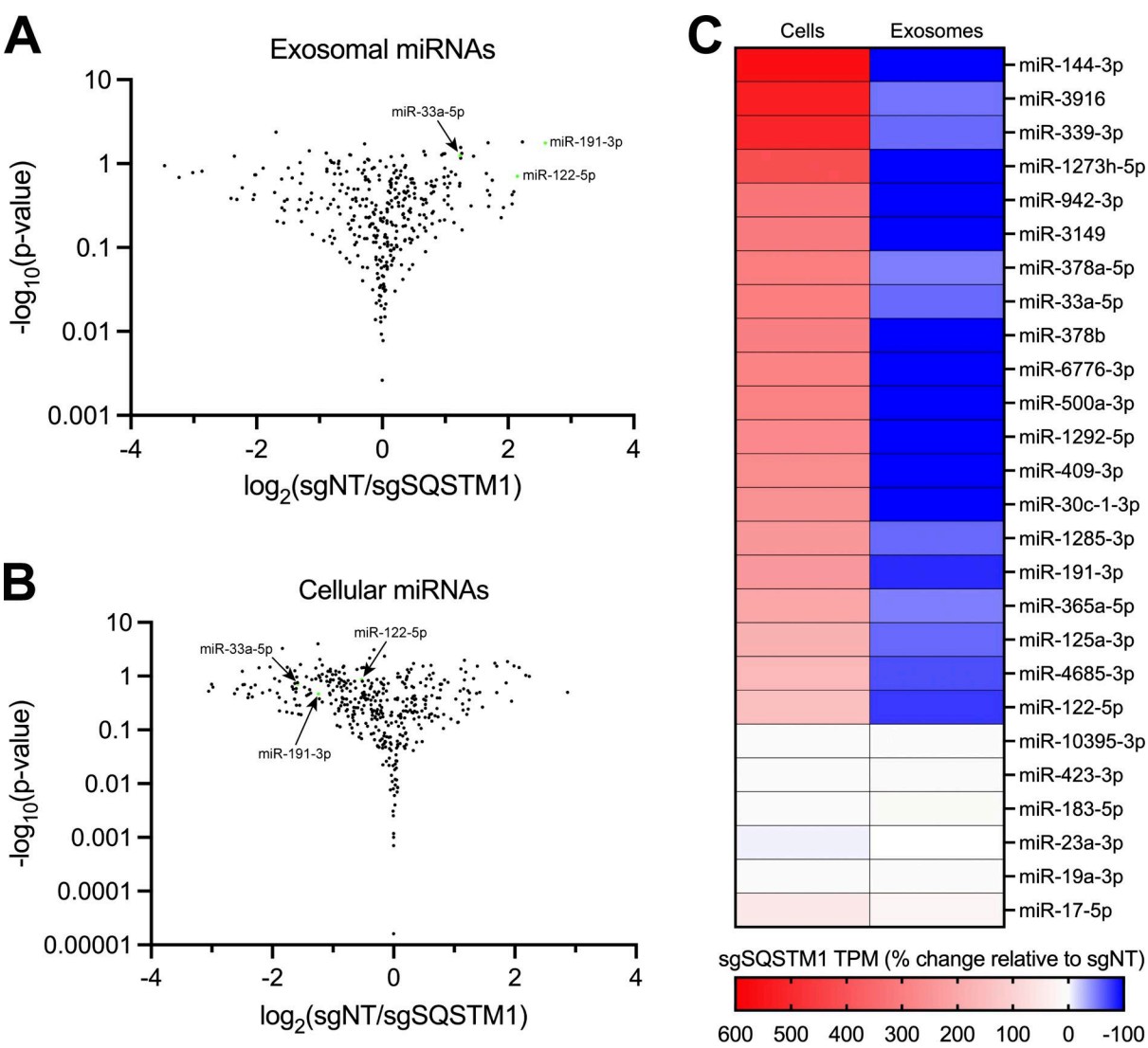

Figure 8. **p62 mediates the secretion of miR-122 and other exosomal miRNAs. (A)** Volcano plot showing miRNAs present in immunoisolated sgNT exosomes compared with sgSQSTM1 exosomes ($n$ = 3). **(B)** Volcano plot showing miRNAs present in HA-CD63-mEGFP[ECL1] sgNT cells compared with sgSQSTM1 cells ($n$ = 3). **(C)** Changes in the intracellular and exosomal levels of miRNAs of interest from HA-CD63-mEGFP[ECL1] sgNT and sgSQSTM1 cells and immunopurified exosomes ($n$ = 3). Mapped miRNA reads were normalized to the total number of miRNA reads in each sample.

3p) in sgSQSTM1 exosomes compared with sgNT exosomes (Fig. 8 A). Comparison of the miRNA profiles between sgNT and sgSQSTM1 cells indicated that these changes were not due to decreased expression of these miRNAs in sgSQSTM1 cells (Fig. 8 B). We then examined the relative content of exosomal miRNA transcripts in relation to their accumulation within cells (Fig. 8 C). We observed that p62 depletion significantly reduced the secretion of many exosomal miRNAs and resulted in their concomitant accumulation within cells. For example, we observed that p62 depletion reduced the secretion of exosomal miR-33a-5p by ∼60% and led to a ∼threefold accumulation of this miRNA within cells. As another example, we noted that p62 depletion completely ablated the secretion of exosomal miR-144-3p and led to a ∼sixfold accumulation of this miRNA inside cells. As a control, we also examined the secretion of p62-independent exosomal miRNAs relative to their intracellular accumulation. We found that p62-independent exosomal miRNAs (such as miR-

17, miR-19a-3p, miR-183-5p, and miR-423-3p) did not accumulate in sgSQSTM1 cells. Thus, we conclude that p62 depletion selectively reduces the secretion of many exosomal miRNAs and results in their accumulation within cells.

## Discussion

Our results demonstrate that p62 mediates the selective capture of La, miR-122, and other miRNAs into the lumen of exosomes derived from the breast cancer cell line MDA-MB-231 (Fig. 9). We propose a model in which ATG7-dependent LC3 lipidation (Fig. 7 D) permits the recruitment of p62 to late endosomes (Fig. 3, E and F), where it interacts with La (Fig. 5) that is bound to miR-122 via LaM and RRM1 (Fig. 6, D and F). p62 then sorts the La and miR-122 RNP into ILVs, and the tripartite complex is secreted within exosomes upon MVB exocytosis (Fig. 4, G–I, Fig. 5, Fig. 7, Fig. 8, and Fig. S5 A). Given the requirement for p62

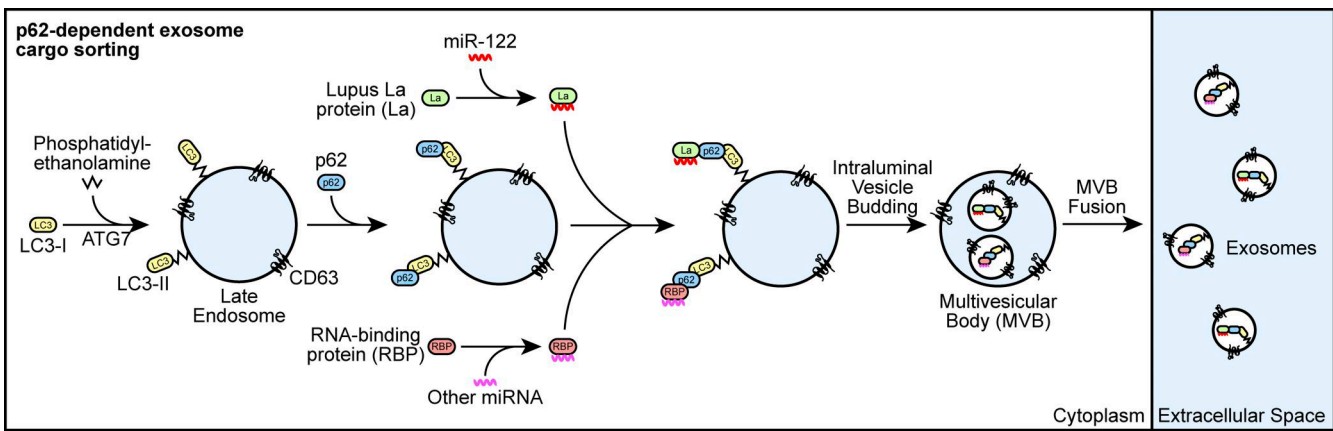

**Figure 9. Schematic depicting the current model of p62-dependent exosome secretion.** Upon ATG7-dependent conjugation of LC3-II to late endosomes, p62 localizes to late endosomes and recruits cytoplasmic RBPs that are bound to selected miRNAs (such as La bound to miR-122 via LaM and RRM1). The RNP complexes are then sequestered into ILVs and secreted within exosomes upon fusion of MVBs with the plasma membrane.

in the secretion of many other exosomal miRNAs (Fig. 8), we speculate that p62 may also capture other RBPs into ILVs.

### Cancer cells exploit p62-dependent exosome cargo sorting to eliminate tumor suppressor miRNAs

Our small RNA-sequencing data demonstrated that many miRNAs that require p62 for their exosomal secretion accumulate in p62-deficient cells (Fig. 8 C). For example, p62 depletion completely blocked the secretion of miR-144 in exosomes and led to a ~sixfold accumulation of this miRNA within cells. We noted that many of the miRNAs that require p62 for their secretion within exosomes and accumulate in p62-deficient cells (including, but not limited to, miR-33a, miR-122, miR-125a, and miR-339) have been implicated in the suppression of tumor cell growth (Guo et al., 2009; Guo et al., 2013; Tsai et al., 2009; Li et al., 2012; Ninio-Many et al., 2013; Liu et al., 2015; Liu et al., 2016; Liu et al., 2019; Pan et al., 2016; Weber et al., 2016; Karatas et al., 2017; Hui et al., 2018; Weihua et al., 2020). As an example, multiple studies have indicated that miR-144 functions as a tumor suppressor miRNA (Guo et al., 2013; Liu et al., 2016; Liu et al., 2019; Pan et al., 2016). In one study, Pan et al. observed that miR-144 expression was significantly downregulated in patient-derived breast cancer tissues compared with patient-matched adjacent normal tissues (Pan et al., 2016). Additionally, the authors showed that miR-144 expression was downregulated in MDA-MB-231 cells compared with the normal breast epithelial cell line Hs578Bst. Interestingly, ectopic miR-144 expression in MDA-MB-231 cells decreased cell proliferation and migration, whereas miR-144 inhibition promoted cell growth.

Other groups have proposed that cancer cells dispose of tumor suppressor miRNAs via EVs to promote tumor cell invasion. In one study, Ostenfeld et al. demonstrated that bladder cancer cells secrete EVs enriched with the tumor-suppressing miRNA miR-23b and proposed that EV-mediated miR-23b clearance promotes tumor cell invasiveness (Ostenfeld et al., 2014). Ostenfeld et al. demonstrated that genetic depletion of Rab27a or Rab27b attenuated miR-23b secretion and resulted in intracellular miR-23b accumulation. Additionally, the authors

performed Matrigel invasion assays and found that Rab27 knockdown or ectopic miR-23b expression inhibited invasion *in vitro*. In other work, Cha et al. proposed that EV-mediated disposal of tumor suppressor miR-100 from cancer cells may serve to promote tumor cell invasion (Cha et al., 2015).

These observations, taken together with our results, lead us to propose that cancer cells exploit p62-dependent exosome cargo sorting to selectively eliminate tumor suppressor miRNAs as a means to promote tumor cell proliferation and invasion. Consistent with this proposal, p62 is aberrantly overexpressed in multiple types of cancer (e.g., breast, kidney, liver, and lung cancer), correlates with poor cancer patient prognosis, and has been demonstrated to promote tumor cell proliferation and tumorigenesis (Thompson et al., 2003; Rolland et al., 2007; Duran et al., 2008; Mathew et al., 2009; Moscat and Diaz-Meco, 2009; Inami et al., 2011; Inoue et al., 2012; Li et al., 2013).

### A secretory autophagy pathway for exosomal La secretion

Our data also contribute to developing appreciation of the dynamic interplay between autophagy and EV secretion. The relationship between autophagy and EV secretion was initially observed in *Saccharomyces cerevisiae* subjected to nitrogen deprivation (Duran et al., 2010). While studying the unconventional secretion of the acyl coenzyme A–binding protein Acb1, Duran et al. observed that Acb1 secretion from *S. cerevisiae* requires components of the autophagy pathway and the endosomal sorting complex required for transport (ESCRT) (Duran et al., 2010). This connection between autophagy and EV secretion was later extended to mammalian cells with the demonstration that serum starvation of human vascular endothelial cells promotes caspase 3–dependent secretion of LC3-II, ATG16L1, and LAMP2 (Sirois et al., 2012; Pallet et al., 2013). Since then, multiple components of the autophagy pathway have been demonstrated to facilitate EV secretion. For example, Murrow et al. identified a complex between ATG12, ATG3, and ALIX that promotes late endosome function and EV secretion (Murrow et al., 2015). Additionally, Guo et al. demonstrated that ATG5 promotes exosome secretion by dissociating the V-ATPase (Guo et al., 2017).

Interestingly, ATG5 dissociates the V-ATPase by sequestering the regulatory component ATP6V1E1 into ILVs.

In a recent key study, Leidal et al. discovered that endosomal LC3-II mediates the selective capture of multiple RBPs into exosomes through a pathway termed LC3-dependent EV loading and secretion (LDELS) (Leidal et al., 2020). The authors demonstrated that endosomal LC3-II directly captures LIR-containing RBPs, such as scaffold-attachment factor B (SAFB), into ILVs. They also present evidence that LDELS requires neutral sphingomyelinase 2 but not canonical autophagy or most ESCRT proteins. Our observations that p62 and LC3-II, but not FIP200, are required for La secretion (Fig. 7) indicate that p62 mediates La secretion through LDELS, as La does not contain a LIR (Jacomin et al., 2016; Chatzichristofi et al., 2023). We note that the secretion of La via LDELS may be context-dependent as a previous report did not detect La within EVs isolated from a lymphoblastoid cell line (Baglio et al., 2016).

The Debnath team also recently reported that lysosome inhibition by BafA1 treatment results in the secretion of sequestered autophagic material from amphisomes, a hybrid organelle formed upon fusion of an autophagosome with a late endosome (Solvik et al., 2022). Solvik et al. termed this pathway secretory autophagy during lysosome inhibition (SALI) and suggested that SALI promotes cellular homeostasis during lysosome dysfunction. We note that Solvik et al. demonstrated that p62 and other selective autophagy cargo receptors (e.g., NBR1 and OPTN) are secreted in a sedimentable, protease-accessible form upon BafA1 treatment (Solvik et al., 2022). In contrast, our proteinase K protection experiments using immunopurified exosomes indicate that exosomal p62 is primarily protease inaccessible in the absence of detergent (Fig. 4, H and I). One explanation for the differing data is that we monitored exosome secretion from unperturbed cells that contain acidic lysosomes (Fig. 4 C). Our data are consistent with those of Leidal et al., who demonstrated that EVs isolated from unperturbed cells contain LC3-II, which is resistant to protease in the absence of detergent (Leidal et al., 2020). Our data indicate that p62 can contribute to LDELS and suggest that proteins without a LIR can be captured by selective autophagy receptors and secreted via LDELS.

Previously, we demonstrated that liquid–liquid phase separation is required to sort YBX1 and miR-223 into exosomes (Liu et al., 2021). Analogously, phase separation has been implicated in the selective capture of cytosolic proteins into autophagosomes (Sun et al., 2018; Zhang et al., 2018; Yamasaki et al., 2020). In one study, Sun et al. demonstrated that p62 phase condensates are required for autophagic cargo sequestration (Sun et al., 2018). Interestingly, we observed that a portion of cytoplasmic La localizes to P-bodies (Fig. S2 A) and suggest that La may localize and be sequestered into ILVs by p62 phase condensates. Consistent with this proposal, LC3 lipidation is required for the clearance of P-body and stress granule components (Buchan et al., 2013). The binding interface between La and p62 remains to be examined.

**EV secretion may serve to promote cellular homeostasis**
Many EV studies suggest, by analogy to membrane-enveloped viruses, that EVs fuse at the plasma membrane or with the late endosome limiting membrane to deliver soluble signaling molecules to the cytoplasm of recipient cells (Meldolesi, 2018). Despite broad interest in EV-mediated intercellular communication, various studies have indicated that the delivery of luminal EV cargo is inefficient (Luhtala and Hunter, 2018; de Jong et al., 2020; Albanese et al., 2021; Somiya and Kuroda, 2021; Zhang and Schekman, 2023). Although we do not rule out the existence of specialized avenues for luminal EV cargo delivery between cells, we speculate that a major function of EV secretion is to promote cellular homeostasis.

Consistent with this suggestion, various studies have demonstrated that EVs are loaded with cellular material that requires elimination. For example, maturing reticulocytes eliminate hundreds of proteins irrelevant to erythrocyte function, such as the transferrin receptor, through secretion in exosomes (Díaz-Varela et al., 2018). Other work has indicated that hyper-stimulated G-protein–coupled receptors are excised from the ends of primary cilia and secreted in microvesicles (Nager et al., 2017). Intriguingly, we note an emerging trend in which nuclear RBPs (including, but not limited to, La, hnRNPA2B1, HuR/ELAVL1, SYNCRIP, SAFB, and hnRNPK) are selectively loaded into EVs, often using machinery associated with degradation such as LC3 and p62 (Villarroya-Beltri et al., 2013; Mukherjee et al., 2016; Santangelo et al., 2016; Temoche-Diaz et al., 2019; Leidal et al., 2020). This enrichment leads us to speculate that mislocalized nuclear RBPs are eliminated by EVs. One explanation as to how nuclear RBPs mislocalize to the cytoplasm is that the nucleoplasm and cytoplasm mix upon disassembly of the nuclear envelope during mitosis (Gerace and Blobel, 1980; Ungricht and Kutay, 2017). We propose that p62 sequesters mislocalized nuclear RBPs into ILVs for elimination upon MVB exocytosis.

As an example of RNA cargo requiring elimination, Koppers-Lalic et al. demonstrated that EVs derived from cultured B cells and human urine samples are enriched with small RNAs that have been posttranscriptionally modified by 3′ non-templated polyuridine addition, an established "mark" for RNA degradation (Li et al., 2005; Heo et al., 2008; Koppers-Lalic et al., 2014; Lim et al., 2014). Interestingly, La has been detected by mass spectrometry in EVs isolated from B cells (Meckes et al., 2013). Given our previous demonstration that La sorts miR-122 into ILVs via its polyuridine tract (Temoche-Diaz et al., 2019), it is tempting to speculate that La may deliver mislocalized and/or damaged RNA transcripts that have been marked for degradation into ILVs to ensure their elimination. If so, the terminal uridyltransferases TUT4 and TUT7 may modulate the RNA composition of exosomes.

Cargo-independent roles for EV-mediated homeostasis have also been proposed. For example, Keller et al. demonstrated that secreted exosomes act as a "decoy" against pore-forming toxins released by bacterial pathogens during infection (Keller et al., 2020). In recent work, we demonstrated that EV secretion is coupled to plasma membrane repair (Williams et al., 2023; Williams et al., 2025). Consistent with our observations, Whitham et al. observed in a cohort of human participants that acute exercise stimulates the release of EV-associated proteins into circulating plasma (Whitham et al., 2018). We propose that

evaluating EVs based on their ability to facilitate cellular homeostasis provides an alternative framework in which to consider the physiological functions of EVs (Ngo et al., 2025).

## Materials and methods

### Cell lines, media, and general chemicals

MDA-MB-231 and HEK293T cell lines were cultured in 5% $CO_2$ at 37°C and maintained in DMEM supplemented with 10% FBS (Thermo Fisher Scientific). The HEK293T ATG7 KO and FIP200 KO cells were kind gifts from Dr. Jayanta Debnath (UC San Francisco, San Francisco, CA, USA) and Dr. Roberto Zoncu (UC Berkeley, Berkeley, CA, USA), respectively.

For the experiments detailed in Fig. 2, the MDA-MB-231 APEX2 and La-APEX2 cells were cultured in DMEM supplemented with 10% dialyzed FBS (Cytiva Life Sciences). For the EV isolation experiments detailed in Fig. 4, F–I; Fig. 5 B; Fig. 7, B and C; and Fig. S5 A, MDA-MB-231 cells were grown in DMEM supplemented with 10% FBS and switched to EV-depleted medium 24 h prior to EV isolation. EV-depleted medium was produced by centrifugation at 186,000 × $g$ (40,000 RPM) for 24 h at 4°C using a Type 45Ti rotor (Beckman Coulter). For the EV isolation experiment detailed in Fig. 7, E and H, HEK293T WT, ATG7 KO, and FIP200 KO cells were grown in DMEM supplemented with 10% FBS and switched to serum-free DMEM 24 h prior to EV isolation. For the small RNA-sequencing experiments detailed in Fig. 8, MDA-MB-231 HA-CD63-mEGFP$^{ECL1}$ sgNT and sgSQSTM1 cells were grown in DMEM supplemented with 10% exosome-depleted FBS (System Biosciences).

Cells were routinely assessed and found negative for mycoplasma contamination using the MycoAlert Mycoplasma Detection Kit (Lonza Biosciences). All of the cell lines used in this study were authenticated by the UC Berkeley Cell Culture facility using STR profiling. Unless otherwise noted, all other chemicals were purchased from Sigma-Aldrich.

### Antibodies

Primary antibodies used for immunoblot (1:1,000 dilution) were mouse anti-ALIX (sc-53540; Santa Cruz), rabbit anti-annexin A2 (ab185957; Abcam), rabbit anti-ATG7 (D12B11; Cell Signaling Technology), mouse anti-CD63 (556019; BD Biosciences), rabbit anti-FIP200 (aka RB1CC1, 17250-1-AP; Proteintech), mouse anti-flotillin 2 (610384; BD Biosciences), rabbit anti-HA-Tag (C29F4; Cell Signaling Technology), mouse anti-La/SSB (OTI1E11; OriGene), rabbit anti-La/SSB (D19B3; Cell Signaling Technology), rabbit anti-LAMP1 (D2D11; Cell Signaling Technology), rabbit anti-LAMP2A (ab18528; Abcam), mouse anti-LC3B (E5Q2K; Cell Signaling Technology), mouse anti-p62/SQSTM1 (ab56416; Abcam), rabbit anti-p62/SQSTM1 (PM045; MBL International), rabbit anti-Rab5 (D4F5; Cell Signaling Technology), mouse anti-TIM23 (611222; BD Biosciences), and rabbit anti-vinculin (ab129002; Abcam). Secondary antibodies used for immunoblot (1:10,000 dilution) were HRP-linked sheep anti-mouse IgG (NA931; Cytiva) and HRP-linked donkey anti-rabbit IgG (NA934; Cytiva).

Primary antibodies used for immunofluorescence (1:100 dilution) were mouse anti-La/SSB (OTI1E11; OriGene), rabbit anti-La/SSB (D19B3; Cell Signaling Technology), mouse anti-CD63 (556019; BD Biosciences), rabbit anti-DDX6 (A300-461A; Bethyl Laboratories), rabbit anti-LC3B (NB100-2220; Novus Biologicals), and guinea pig anti-p62/SQSTM1 (GP62-C; Biogen Biotechnik). Secondary antibodies used for immunofluorescence (1:500 dilution) were Alexa Fluor 488 donkey anti-rabbit IgG (A-21202; Invitrogen), Alexa Fluor 555 goat anti-mouse IgG (A-21424; Invitrogen), and Alexa Fluor 647 goat anti-guinea pig IgG (A-21450; Invitrogen).

Antibodies used for immunoprecipitation (5 μg per IP) were mouse mAb IgG1 Isotype Control (G3A1; Cell Signaling Technology) and mouse anti-La/SSB (OTI1E11; OriGene).

### Plasmid constructs

HA-CD63-mEGFP$^{ECL1}$ was cloned into a pLJM1-Blast backbone. An HA tag was appended at the N terminus of CD63, and mEGFP, flanked at both sides by GGGGS linkers, was inserted into the first extracellular loop between Gln36 and Leu37. The all-in-one CRISPRi plasmids (lenti-UCOE-hU6-sgRNA-EF1α-Zim3-KRAB-dCas9-P2A-PuroR) were cloned into a lentiCRISPR-v2 backbone. Constructs for recombinant purification of full-length and truncated La were cloned into a pET28a backbone. The construct for recombinant purification of full-length p62 (pET28a-His6-MBP-p62) was by Ingersoll et al. (2025). All recombinant La proteins were appended at the C terminus with nanoluciferase (Nluc), a FLAG tag, and a His$_6$ tag. All plasmid constructs were verified by whole-plasmid sequencing (Plasmidsaurus).

### Lentivirus production and transduction

HEK293T cells at ~40% confluence in a 6-well plate were transfected with 165 ng of pMD2.G, 1.35 μg of psPAX2, and 1.5 μg of a lentiviral plasmid using the TransIT-LT1 Transfection Reagent (Mirus Bio) according to the manufacturer's protocol. The lentivirus-containing medium was harvested 48 h after transfection by filtration through a 0.45-μm polyethersulfone filter (VWR Sciences). The filtered lentivirus was aliquoted, snap-frozen in liquid nitrogen, and stored at –80°C. MDA-MB-231 cells were transduced with filtered lentivirus in the presence of 8 μg/ml polybrene for 24 h, and then the medium was replaced. Cells were selected using 1 μg/ml puromycin or 5 μg/ml blasticidin S for 3 and 7 days, respectively.

### CRISPRi

The CRISPRi protospacer sequence targeting the promoter region of SQSTM1 was selected using the CRISPick tool (Broad Institute) and cloned into lenti-UCOE-hU6-sgRNA-EF1α-Zim3-KRAB-dCas9-P2A-PuroR. The nontargeting control protospacer sequence used was 5′-AAAACAGGACGATGTGCGGC-3′, and the SQSTM1-targeting protospacer sequence used was 5′-GTGAGCGACGCCATAGCGAG-3′. The CRISPRi lentiviruses were used to transduce the parental MDA-MB-231 HA-CD63-mEGFP$^{ECL1}$ cells. The transduced cells were isolated by puromycin selection and utilized for downstream experiments.

### Immunoblotting

Cells were lysed in PBS supplemented with 1% TX-100 and a protease inhibitor cocktail (1 mM 4-aminobenzamidine

dihydrochloride, 1 µg/ml antipain dihydrochloride, 1 µg/ml aprotinin, 1 µg/ml leupeptin, 1 µg/ml chymostatin, 1 mM PMSF, 50 µM N-tosyl-L-phenylalanine chloromethyl ketone, and 1 µg/ml pepstatin) and incubated on ice for 15 min. The whole-cell lysate was centrifuged at 15,000 × $g$ for 10 min at 4°C, and the PNS was diluted with 6X Laemmli buffer (nonreducing) to a 1X final concentration. Samples were heated at 95°C for 5 min and resolved in a 4–20% acrylamide Tris-glycine gradient gels (Life Technologies). The resolved proteins were then transferred to polyvinylidene difluoride (PVDF) membranes (EMD Millipore).

For immunoblots, the PVDF membranes were blocked for 30 min with 5% BSA in TBS supplemented with 0.1% Tween-20 (TBS-T) and incubated overnight with 1:1,000 dilutions of primary antibodies in 5% BSA in TBS-T. The membranes were then washed three times with TBS-T, incubated for 1 h at room temperature with 1:10,000 dilutions of HRP-conjugated secondary antibodies (Cytiva Life Sciences) in 5% BSA in TBS-T, washed three times with TBS-T, washed once with TBS, and then detected with ECL2 or PicoPLUS reagent (Thermo Fisher Scientific).

### Immunofluorescence and live-cell microscopy

For immunofluorescence experiments, MDA-MB-231 WT cells grown to ~70% confluence on 12 mm poly-D-lysine–coated coverslips (Corning) were washed once with PBS, fixed in 4% EM-grade paraformaldehyde (Electron Microscopy Science) for 20 min at room temperature, washed three times with PBS, and permeabilized/blocked in either saponin blocking buffer (2% FBS and 0.02% saponin in PBS) or TX-100 blocking buffer (2% FBS and 0.1% TX-100 in PBS) for 30 min at room temperature. Cells were then placed in a humidity chamber, incubated with a 1:100 dilution of primary antibody in blocking buffer for 1 h at room temperature, washed three times with PBS, incubated with a 1:500 dilution of fluorophore-conjugated secondary antibody in blocking buffer for 1 h at room temperature, and washed three times with PBS. The coverslips were then mounted overnight in ProLong Diamond antifade mountant with DAPI (Thermo Fisher Scientific) and sealed with clear nail polish before imaging. Images were acquired using an LSM900 confocal microscope system (ZEISS) using Airyscan 2 mode and a 63X Plan-Apochromat, NA 1.40 objective.

For live-cell imaging, MDA-MB-231 HA-CD63-mEGFP[ECL1] cells were treated with either DMSO or 100 nM BafA1 (Cayman Chemical) for 16 h and then labeled with LysoTracker DND-99 Red (Invitrogen) according to the manufacturer's protocol. Images were acquired using an Echo Revolve microscope system (Echo) using the 10X, NA 0.40 objective.

### Membrane fractionation

MDA-MB-231 WT cells were grown to ~90% confluence in 3 × 150-mm dishes. All subsequent manipulations were performed at 4°C. Each 150-mm dish was washed once with 5 ml of cold PBS and then harvested by scraping into 5 ml of cold PBS. The cells were collected by centrifugation at 300 × $g$ for 5 min, and the supernatant was discarded. The cell pellets were resuspended in 2 vol of cold homogenization buffer (HB) (20 mM HEPES, pH 7.4, 250 mM D-sorbitol, 1 mM EDTA, and a protease inhibitor

cocktail [see immunoblotting section]). After 10 min, the cell suspension was mechanically lysed by passing through a 22-gauge needle until ~85% of cells were lysed as assessed by trypan blue exclusion. The lysed cells were centrifuged at 1,000 × $g$ for 15 min to sediment intact cells and nuclei, and the PNS was transferred to a separate tube.

For the membrane fractionation experiments detailed in Fig. 1 A and Fig. 3 A, the PNS was subjected to sequential differential centrifugation at 5,000 × $g$ (15 min), 20,000 × $g$ (20 min), and ~100,000 × $g$ (30 min, TLA-55 rotor) to collect the membranes sedimented at each speed. The isolated membrane pellet fractions were resuspended in equal volumes of 1X Laemmli buffer (nonreducing) for immunoblot analysis.

For the proteinase K protection assays detailed in Fig. 1 C and Fig. 3 C, the PNS was centrifuged at 20,000 × $g$ for 20 min to collect a 20,000 × $g$ membrane fraction. The sedimented membranes were resuspended in 200 µl of cold HB (without protease inhibitors) and distributed to five equal aliquots. The samples were left untreated, treated with 20 or 40 µg/ml proteinase K (New England Biolabs), or mixed with TX-100 (1% final) for 5 min on ice prior to treatment with 20 or 40 µg/ml proteinase K. The reactions were incubated on ice for 20 min, followed by proteinase K inactivation using 5 mM PMSF (5 min incubation on ice). The samples were then mixed with 6X Laemmli buffer (nonreducing) to a 1X final concentration for immunoblot analysis.

### Unbiased proximity biotinylation and mass spectrometry

MDA-MB-231 cells expressing APEX2 or La-APEX2 were grown to 90% confluency in 4 × 150-mm dishes. The cells were incubated with DMEM supplemented with 10% dialyzed FBS and 500 µM biotin phenol (Iris Biotech) for 30 min, and then $H_2O_2$ was added (1 mM final concentration) for 30 s. All subsequent manipulations were performed at 4°C. The labeling reactions were quenched with quencher solution (10 mM sodium ascorbate, 5 mM Trolox, and 10 mM sodium azide in PBS). Each 150-mm dish was washed three times with 5 ml of cold quencher solution and harvested by scraping into 5 ml of cold PBS. The cells were collected by centrifugation at 300 × $g$ for 5 min, and the supernatant was discarded. The cell pellets were then resuspended in 2 volumes of cold HB. After 10 min, the cell suspension was mechanically lysed by passing through a 22-gauge needle until ~85% of cells were lysed as assessed by trypan blue exclusion. The lysed cells were centrifuged at 1,000 × $g$ for 15 min to sediment intact cells and nuclei, and the PNS fractions were transferred to separate tubes. The PNS fractions were then centrifuged at 20,000 × $g$ for 20 min to obtain 20,000 × $g$ membrane fractions.

The 20,000 × $g$ membrane fractions were each lysed with 500 µl of cold RIPA buffer (50 mM Tris-HCl, pH 7.5, 150 mM NaCl, 0.1% sodium dodecyl sulfate, 0.5% sodium deoxycholate, 1% TX-100, and a protease inhibitor cocktail [see immunoblotting section]) and incubated on a rotating mixer with 100 µl of pre-equilibrated Dynabeads MyOne Streptavidin T1 beads (Invitrogen) overnight. The magnetic beads were then washed as follows: twice with RIPA buffer, once with 1 M KCl, once with 0.1 M $Na_2CO_3$, once with 2 M urea in 20 mM HEPES, pH 8.0, and

twice with RIPA buffer. The bound proteins were then eluted into biotin elution buffer (2X Laemmli buffer supplemented with 10 mM DTT and 2 mM biotin) by heating at 95°C for 10 min.

The eluted proteins were electrophoresed in a 4–20% acrylamide Tris-glycine gradient gel for 3 min at 200 V. The proteins were stained with Coomassie, and the stained bands were excised from the gel using fresh razor blades. The samples were submitted to the Taplin Mass Spectrometry Facility at Harvard University for in-gel tryptic digestion of proteins followed by liquid chromatography and mass spectrometry analysis according to their standards. Proteins enriched in the La-APEX2 sample were then submitted for GO cellular component analysis using the GoTermFinder developed at the Lewis-Sigler Institute at Princeton (Boyle et al., 2004), followed by REVIGO analysis (Supek et al., 2011) and visualization using Cytoscape (Shannon et al., 2003). The mass spectrometry results were then evaluated manually for proteins involved in cargo recognition.

## Organelle immunoprecipitation
MDA-MB-231 HA-CD63-mEGFP[ECL1] cells were grown to ∼90% confluence in 2 × 150-mm dishes. Both 150-mm dishes were washed once with 5 ml of cold PBS, then harvested by scraping into 5 ml of cold PBS. The cells were collected by centrifugation at 300 × $g$ for 5 min at 4°C, and the supernatant was discarded. The cell pellets were resuspended in 2 volumes of cold HB. After 10 min on ice, the cell suspension was mechanically lysed by passing through a 22-gauge needle until ∼85% of cells were lysed as assessed by trypan blue exclusion. The lysed cells were centrifuged at 1,000 × $g$ for 15 min at 4°C to sediment intact cells and nuclei, and the PNS was transferred to a separate tube on ice. Cold HB was added to bring the volume of the PNS up to 1,050 µl. An aliquot (50 µl) of the supernatant fraction was retained for an input measurement, and the remaining lysate was distributed evenly to two tubes. An aliquot (50 µl) of pre-equilibrated anti-HA magnetic beads (Pierce) was added to tube #1, and 50 µl of pre-equilibrated GFP-Trap Magnetic Particles (M-270) (ChromoTek) was added to tube #2. The tubes were then incubated on a rotating mixer for 15 min at room temperature. The magnetic beads were washed three times with cold HB, and the bound material was eluted with 50 µl of cold HB supplemented with 1% TX-100. The eluates were transferred to separate tubes, then mixed with 6X Laemmli buffer (nonreducing) to a 1X final concentration for immunoblot analysis.

## EV and exosome purification
Conditioned medium (420 ml) was harvested from MDA-MB-231 WT, MDA-MB-231 HA-CD63-mEGFP[ECL1], or HEK293T ATG7 KO cells. All subsequent manipulations were performed at 4°C. Cells and large debris were removed by low-speed sedimentation at 1,000 × $g$ for 20 min followed by medium-speed sedimentation at 10,000 × $g$ for 20 min using a fixed-angle FIBERlite F10-6x500y rotor (Thermo Fisher Scientific). The supernatant fraction was then centrifuged at 29,500 RPM for 1.5 h in an SW32 rotor (Beckman Coulter). The pellet fractions were then resuspended in 200 µl of PBS, pooled, and centrifuged at 36,500 RPM for 1 h in an SW55 rotor (Beckman Coulter). The pellet was resuspended in 200 µl of PBS to obtain the final HSP

fraction that was utilized for immunoblot analysis, immunoisolation, or RNA purification.

For exosome immunoisolation, the HSP fraction was diluted with PBS to a final volume of 500 µl and incubated on a rotating mixer with 50 µl of pre-equilibrated GFP-Trap Magnetic Particles (M-270) overnight. The magnetic beads were washed three with PBS and processed for immunoblot, proteinase K protection, IP analysis, or RNA purification.

For the proteinase K protection assays detailed in Fig. 4 H, the anti-GFP nanobody-bound exosomes were distributed to four equal aliquots. Tube #1 was left untreated, tube #2 was spiked with 2 mg of TAMRA-labeled α-synuclein, tube #3 was spiked with 2 mg of TAMRA-labeled α-synuclein and treated with 10 µg/ml proteinase K, and tube #4 was spiked with 2 mg of TAMRA-labeled α-synuclein and mixed with TX-100 (1% final) for 5 min on ice prior to treatment with 10 µg/ml proteinase K. The reactions were incubated on ice for 20 min, followed by proteinase K inactivation using 5 mM PMSF (5-min incubation on ice). The samples were then mixed with 6X Laemmli buffer (nonreducing) to a 1X final concentration for immunoblot analysis.

## Co-immunoprecipitation of La from cells
MDA-MB-231 WT cells were grown to ∼90% confluence in 2 × 150-mm dishes. All subsequent manipulations were performed at 4°C. Each 150-mm dish was washed once with 5 ml of cold PBS, then harvested by scraping into 5 ml of cold PBS. The cells were collected by centrifugation at 300 × $g$ for 5 min, and the supernatant was discarded. The cell pellets were resuspended in 1,050 µl of cold IP buffer (20 mM HEPES, pH 7.4, 100 mM KCl, 5 mM MgCl$_2$, 1% TX-100, and a protease inhibitor cocktail [see immunoblotting section]). After 10 min, the lysed cells were centrifuged at 15,000 × $g$ for 10 min to sediment insoluble material, and the supernatant was transferred to a separate tube. An aliquot (50 µl) of the supernatant fraction was retained for an input measurement, and the remaining lysate was distributed evenly to two tubes. An aliquot (5 µg) of mouse IgG isotype control (G3A1; Cell Signaling Technology) was added to tube #1, and 5 µg of mouse anti-La antibody (OTI1E11; OriGene) was added to tube #2. 50 µl of pre-equilibrated Protein A/G agarose beads (Santa Cruz Biotechnology) was added to each tube, and the mixtures were incubated on a rotating mixer overnight. The beads were then washed three with IP buffer and eluted in 1X Laemmli buffer (nonreducing) for immunoblot analysis.

## Co-immunoprecipitation of La from exosomes
Exosomes were immunoisolated from MDA-MB-231 HA-mEGFP[ECL1] cells using GFP-Trap magnetic particles (M-270) as described above. All subsequent manipulations were performed at 4°C. The immunopurified exosomes were lysed in 1050 µl of cold IP buffer, and 50 µl was retained for an input measurement. The remaining material was distributed evenly to two tubes. An aliquot (5 µg) of mouse IgG isotype control (G3A1; Cell Signaling Technology) was added to tube #1, and 5 µg of mouse anti-La antibody (OTI1E11; OriGene) was added to tube #2. Pre-equilibrated Protein A/G agarose beads (50 µl) were added to each tube, and the mixtures were incubated on a rotating mixer

overnight. The beads were washed three times with IP buffer and eluted in 1X Laemmli buffer (nonreducing) for immunoblot analysis.

## Recombinant protein purification of full-length and truncated La

Recombinant full-length and truncated La proteins were expressed in *Escherichia coli* Rosetta2(DE3)pLysS cells. Pre-cultures (2.5 ml) were grown overnight at 37°C and diluted to 250 ml cultures. The cultures were incubated at 37°C until the $OD_{600}$ reached ~0.6, and protein expression was induced upon addition of 50 μM IPTG for 16 h at 18°C. The cells were harvested by centrifugation at 5,000 × *g* for 20 min at 4°C in a fixed-angle FIBERlite F14-6x250y rotor, and the cell pellets were stored at –80°C until use. All subsequent manipulations were performed at 4°C. The cell pellets were thawed on ice, resuspended in 20 ml of cold Ni-NTA lysis buffer (20 mM Tris-HCl, pH 8.0, 300 mM NaCl, 10 mM imidazole, 2 mM $MgCl_2$, and a protease inhibitor cocktail [see immunoblotting section]), and lysed by sonication. Each lysate was clarified by centrifugation at 20,000 × *g* for 20 min in a fixed-angle FIBERlite F21-8x50y rotor, and the supernatant fractions were incubated in batch on a rotating mixer with ~1 ml of equilibrated HisPur Ni-NTA Resin (Thermo Fisher Scientific) for 2 h. Each resin portion was transferred to gravity flow columns, washed with three column volumes of cold Ni-NTA wash buffer (similar recipe as Ni-NTA lysis buffer but with 50 mM imidazole), and eluted with cold Ni-NTA elution buffer (similar recipe as Ni-NTA lysis buffer but with 250 mM imidazole).

Eluted full-length and truncated La proteins were incubated on a rotating mixer for 2 h with Pierce Anti-FLAG M2 Affinity Gel (Millipore Sigma) pre-equilibrated in TBS and washed three times with TBS. Aliquots of anti-FLAG–bound La proteins were transferred to separate tubes and lysed with 1X Laemmli buffer (reducing) for SDS-PAGE analysis. The remaining anti-FLAG–bound La proteins were eluted with TBS supplemented with 250 μg/ml 3xFLAG peptide (APExBIO). The eluted proteins were applied to Bio-Spin 6 Tris Buffer size-exclusion columns (Bio-Rad) to remove 3xFLAG peptides according to the manufacturer's protocol. The purified La proteins were distributed in aliquots, snap-frozen in liquid nitrogen, and stored at –80°C until use.

## Recombinant protein purification of p62

Recombinant p62 was expressed in *E. coli* Rosetta2(DE3)pLysS cells. A pre-culture (5 ml) was grown overnight at 37°C and diluted to a 500 ml culture. The culture was incubated at 37°C until the $OD_{600}$ reached ~0.6, and protein expression was induced upon addition of 0.5 mM IPTG for 24 h at 16°C. The protein was then purified using a single step Ni-NTA protocol as described above. Recombinant p62 was never frozen and used for immunoprecipitation experiments immediately after purification.

## In vitro–binding assays between recombinant La and p62

50 μl of anti-DYKDDDK Fab-Trap agarose beads (ChromoTek) were washed once with TBS, distributed evenly to two tubes, and incubated with either 3xFLAG peptide or recombinant La-

Nluc-FLAG-$His_6$ for 30 min at room temperature. The beads were washed three times with TBS and incubated with recombinant $His_6$-MBP-p62 for 1 h at room temperature. The beads were then washed three times with TBS and eluted in 1X Laemmli buffer (nonreducing) for immunoblot analysis.

## EMSAs

Fluorescently labeled miRNAs (5′ IRD800) were obtained from Integrated DNA Technologies. EMSA reactions (20 μl) were assembled on ice as follows: 1 nM of fluorescently labeled miRNA was mixed with increasing amounts of purified La (ranging from 61.25 pM–2 μM) in EMSA reaction buffer (25 mM Tris-HCl, pH 8.0, 100 mM KCl, 1.5 mM $MgCl_2$, 1 mM DTT, 0.05% Nonidet P-40, 5% glycerol, 50 μg/ml heparin, and 1 U/μl RiboLock RNase Inhibitor). Each reaction was incubated at 30°C for 30 min, 4°C for 10 min and then mixed with 6X loading buffer (60 mM KCl, 10 mM Tris, pH 7.6, 50% glycerol, and 0.03% [wt/vol] xylene cyanol) to a final 1X concentration. Aliquots of each reaction (5 μl) were resolved in 6% Novex Tris-borate-EDTA polyacrylamide gels (Thermo Fisher Scientific) at 200 V for 30 min in a 4°C cold room, and in-gel fluorescence was detected using an Odyssey CLx Imaging System (LI-COR Biosciences). Fluorescence was quantified using the Image Studio software of the Odyssey CLx Imaging System, and the quantified data points were fit to Hill equations to calculate $K_d$s. The fraction of bound miRNA was calculated as a function of exhaustion of unbound miRNA.

For the competition EMSAs detailed in Fig. S4, $La^{1–334}$ or $La^{1–224}$ was preincubated with fluorescently labeled miR-122 on ice for 5 min prior to mixing with increasing amounts of either $La^{335–408}$ or $La^{225–408}$ in EMSA reaction buffer. The EMSA reactions were then processed as described above.

## RNA purification and quantitative real-time PCR

Extracellular HSP fractions were isolated from MDA-MB-231 HA-CD63-mEGFP$^{ECL1}$ sgNT and sgSQSTM1 cells as described above. TRI reagent (Zymo Research) was added to each HSP fraction, and the RNA was extracted using the Direct-Zol RNA miniprep kit (Zymo Research) with on-column DNase treatment according to the manufacturer's instructions. The purified RNA was then quantified using an Agilent 2100 Bioanalyzer (Agilent Technologies) at the UC Berkeley Functional Genomics Lab. The TaqMan miRNA assay ID for hsa-miR-122-5p was 477855_mir (Life Technologies). As there are no well-accepted endogenous control transcripts for EVs, the relative quantification was normalized to equal amounts of starting RNA material. RNA (1 ng) was reverse transcribed using the TaqMan Advanced miRNA cDNA Synthesis Kit (Applied Biosystems) according to the manufacturer's instructions. Quantitative real-time PCR was performed using a QuantStudio5 real-time PCR system (Applied Biosystems) using TaqMan Fast Advanced Master Mix for qPCR with no UNG (Life Technologies). Relative quantification was calculated from the expression 2^-($C_t$(control)-$C_t$(experimental)).

## RNA purification and small RNA library preparation

MDA-MB-231 HA-CD63-mEGFP$^{ECL1}$ sgNT and sgSQSTM1 cells and immunopurified exosomes were isolated as described above (*n* = 3). TRI reagent was added to each sample, and the RNA was

extracted using the Direct-Zol RNA miniprep kit with on-column DNase treatment according to the manufacturer's instructions. The purified RNA was then quantified using an Agilent 2100 Bioanalyzer. Small RNA libraries were then generated using the SMARTer smRNA-Seq Kit for Illumina (Takara Bio). Input RNA (1 ng of exosomal RNA or 50 ng cellular RNA) from three biological replicates was polyadenylated and reverse transcribed using the PrimeScript reverse transcriptase and oligo (dT) and SMART smRNA oligonucleotides. Primers containing Illumina i5 and i7 adapter sequences were then used to amplify each cDNA sample. The libraries were then purified using the NucleoSpin Gel and PCR Clean-up kit (Takara Bio) and quantified using an Agilent 2100 Bioanalyzer. The libraries were sent to Azenta Life Sciences for sequencing on a NovaSeq 6000 sequencer.

### RNA-sequencing analysis

The raw sequencing reads were trimmed using cutadapt 4.9 (Martin, 2011) with the following script: cutadapt -m 15 -u 3 -a AAAAAAAAAA -j 0 -o /path/to/output/fastq.gz /path/to/input/fastq.gz. The trimmed reads were then mapped to human miR-Base 22.1 using miRDeep2 with the following script: miRDeep2.pl /path/to/collapsedReads.fa /path/to/genome.fa /path/to/mappedReads.arf /path/to/mature-miRNAs.fa /path/to/hairpin-miRNAs.fa -t Human 2>report.log. Mapped miRNA reads were normalized by dividing the reads per miRNA by the total number of miRNA reads per sample, and this value was multiple by one million (transcripts per million miRNA mapped transcripts). P values were calculated using an unpaired two-tailed $t$ test.

### Online supplemental material

Fig. S1 shows the additional Airyscan immunofluorescence images. Fig. S2 shows the co-localization of La with the P-body marker, DDX6. Fig. S3 shows the purification of recombinant proteins. Fig. S4 shows the competition EMSAs between truncated La proteins. Fig. S5 shows the p62 is required for the secretion of miR-122. Table S1 shows the APEX2 proximity labeling data from Fig. 2.

### Data availability

The APEX2 dataset is available in the supplementary material. RNA-sequencing data have been deposited to the NIH SRA under accession code PRJNA1377456.

## Acknowledgments

We dedicate this work to Bob Lesch, our lab manager for the past several decades, who was tragically taken from us by an accident in 2021. We would like to thank all members of the Schekman lab for fruitful discussions during the preparation of this manuscript. We would also like to thank our current lab manager, Nam Che; Ross Tomaino from the Harvard Taplin Mass Spectrometry Facility; Alison Killilea from the UC Berkeley Cell Culture Facility (SCR_017924); and Justin Choi from the UC Berkeley Functional Genomics Lab (SCR_022170). The ATG7 and FIP200 knockout cell lines were generous gifts from Jayanta

Debnath (UC San Francisco, San Francisco, CA, USA) and Roberto Zoncu (UC Berkeley, Berkeley, CA, USA), respectively. The plasmid for purification of recombinant p62 and the TAMRA-labeled α-synuclein protein were kind gifts from Michael Rape (UC Berkeley, Berkeley, CA, USA) and Shenjie Wu (UC Berkeley, Berkeley, CA, USA), respectively.

J.M. Ngo was supported by an NSF Graduate Research Fellowship and a Ruth L. Kirschstein NRSA Predoctoral Fellowship (F31CA284881). R. Schekman is an Investigator of the Howard Hughes Medical Institute, a Senior Fellow of the UC Berkeley Miller Institute of Science, and Chair of the Scientific Advisory Board of Aligning Science Across Parkinson's Disease. This work was funded by the Howard Hughes Medical Institute. The funders had no role in study design, data collection and interpretation, or the decision to submit the work for publication.

Author contributions: Jordan Matthew Ngo: conceptualization, data curation, formal analysis, investigation, methodology, software, supervision, validation, visualization, and writing—original draft, review, and editing. Justin Krish Williams: conceptualization, methodology, and writing—review and editing. Morayma Mercedes Temoche-Diaz: conceptualization, formal analysis, investigation, methodology, and writing—review and editing. Abinayaa Murugupandiyan: investigation. Randy Schekman: conceptualization, funding acquisition, project administration, supervision, and writing—review and editing.

Disclosures: The authors declare no competing interests exist.

Submitted: 20 March 2025

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

# Supplemental material

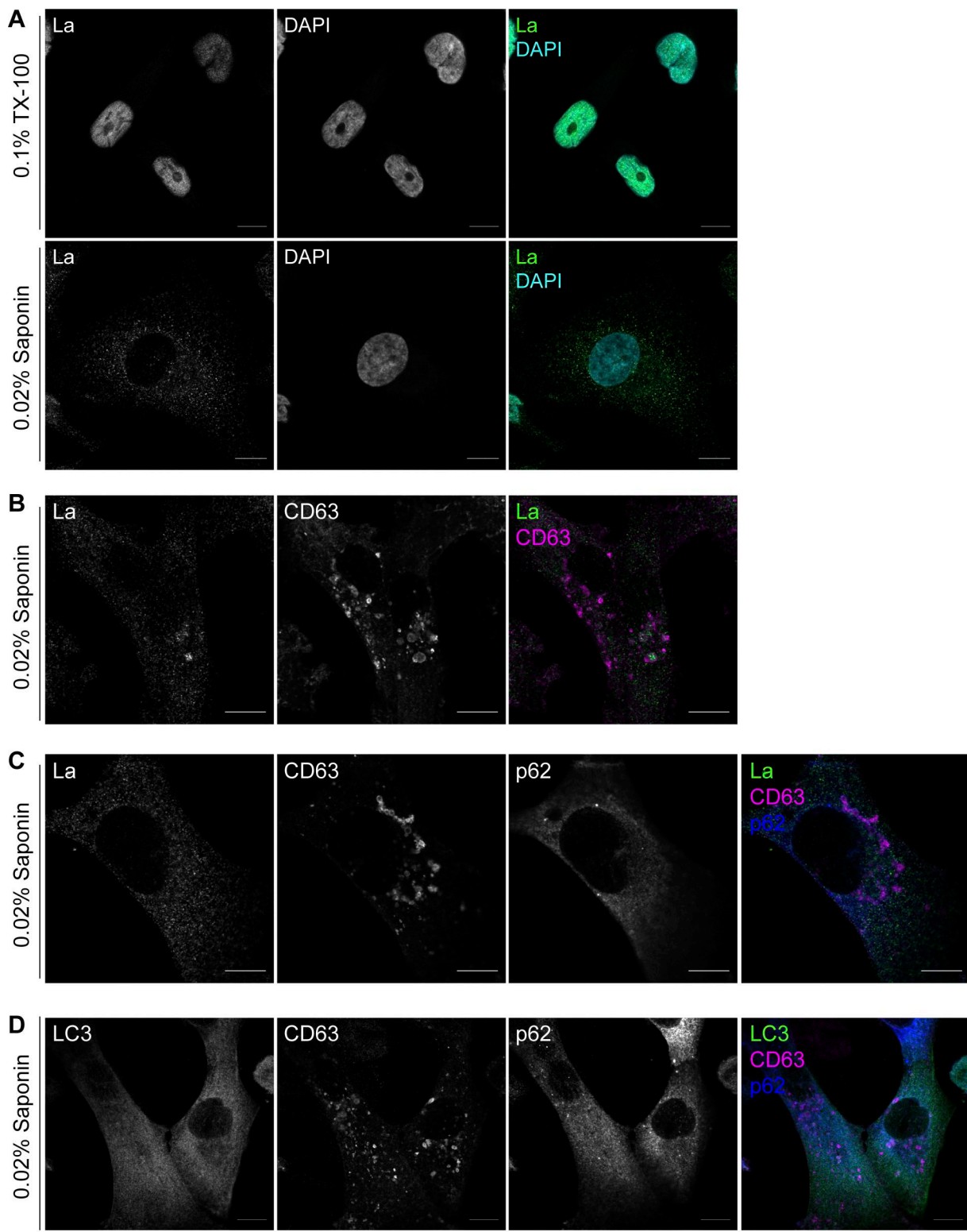

**Figure S1. Additional Airyscan immunofluorescence images. (A)** Airyscan microscopy of endogenous La with DAPI counterstain from MDA-MB-231 WT cells permeabilized with either 0.1% TX-100 or 0.02% saponin. Green: La; cyan: DAPI. Scale bar: 10 µm. **(B)** Airyscan microscopy of endogenous La and CD63 from MDA-MB-231 WT cells permeabilized with 0.02% saponin. Green: La; magenta: CD63. Scale bar: 10 µm. **(C)** Airyscan microscopy of endogenous La, CD63, and p62 from MDA-MB-231 cells permeabilized with 0.02% saponin. Green: La; magenta: CD63; blue: p62. Scale bar: 10 µm. **(D)** Airyscan microscopy of endogenous LC3, CD63, and p62 from MDA-MB-231 cells permeabilized with 0.02% saponin. Green: LC3; magenta: CD63; blue: p62. Scale bar: 10 µm.

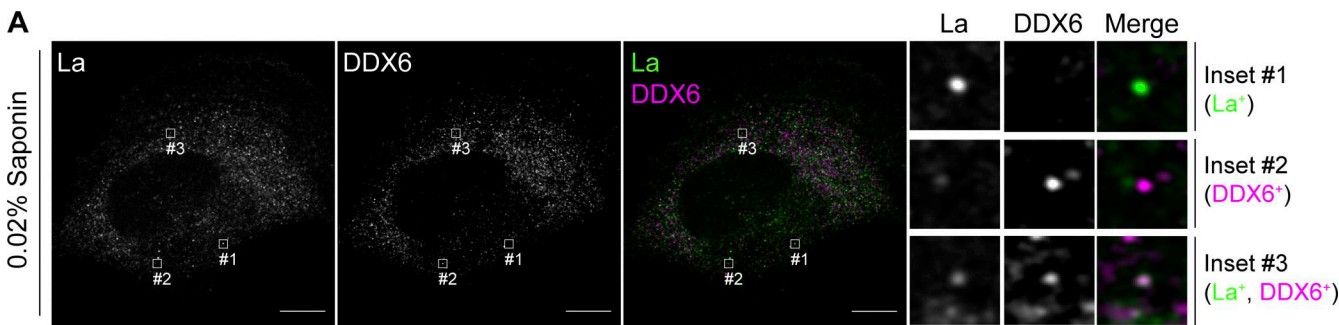

**Figure S2. Co-localization of La with the P-body marker, DDX6. (A)** Airyscan microscopy of endogenous La and DDX6 from MDA-MB-231 WT cells permeabilized with 0.02% saponin. Insets indicate La-positive, DDX6-positive, and La/DDX6 double-positive puncta. Green: La; magenta: DDX6. Scale bar: 10 μm.

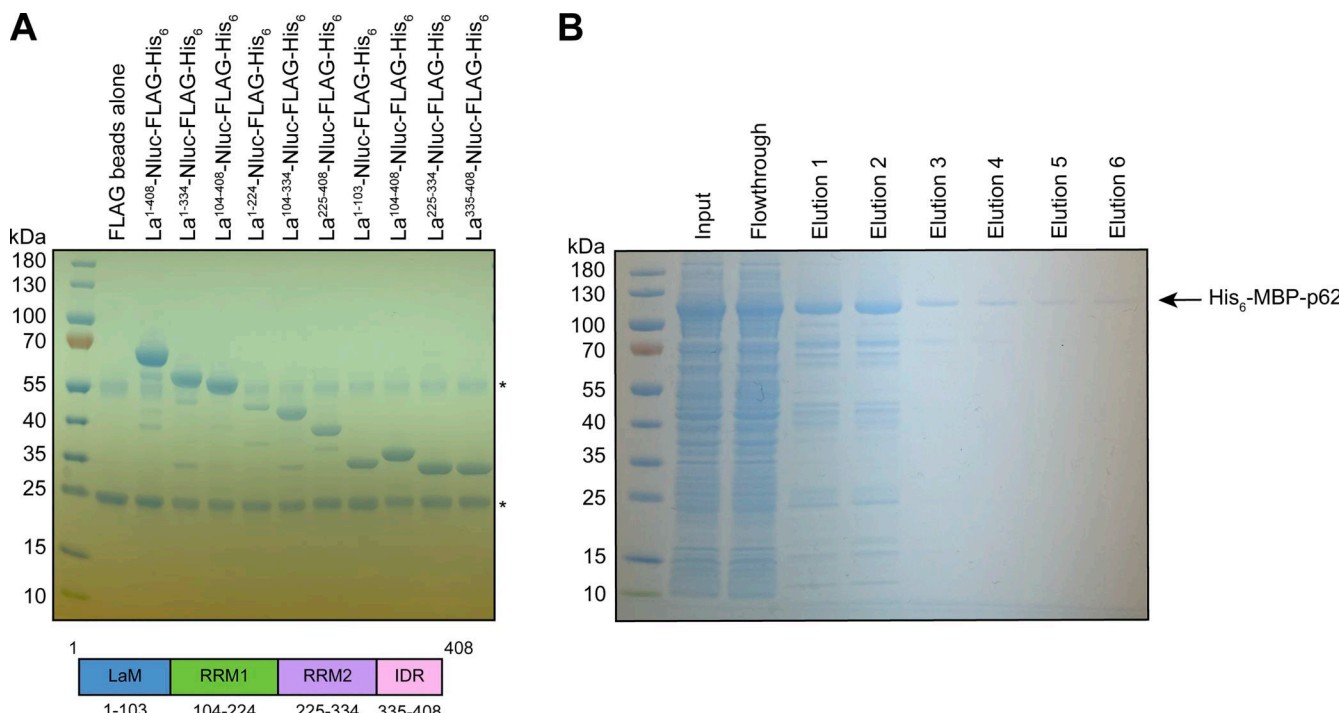

**Figure S3. Recombinant protein purification. (A)** Coomassie-stained SDS-PAGE gel showing anti-FLAG–bound full-length and truncated La proteins prior to elution using 3xFLAG peptide. Asterisks (*) indicate anti-FLAG antibody fragments that were eluted from the resin upon denaturing elution with 1X Laemmli buffer (reducing). **(B)** Coomassie-stained SDS-PAGE gel showing the purification of recombinant p62. Source data are available for this figure: SourceData FS3.

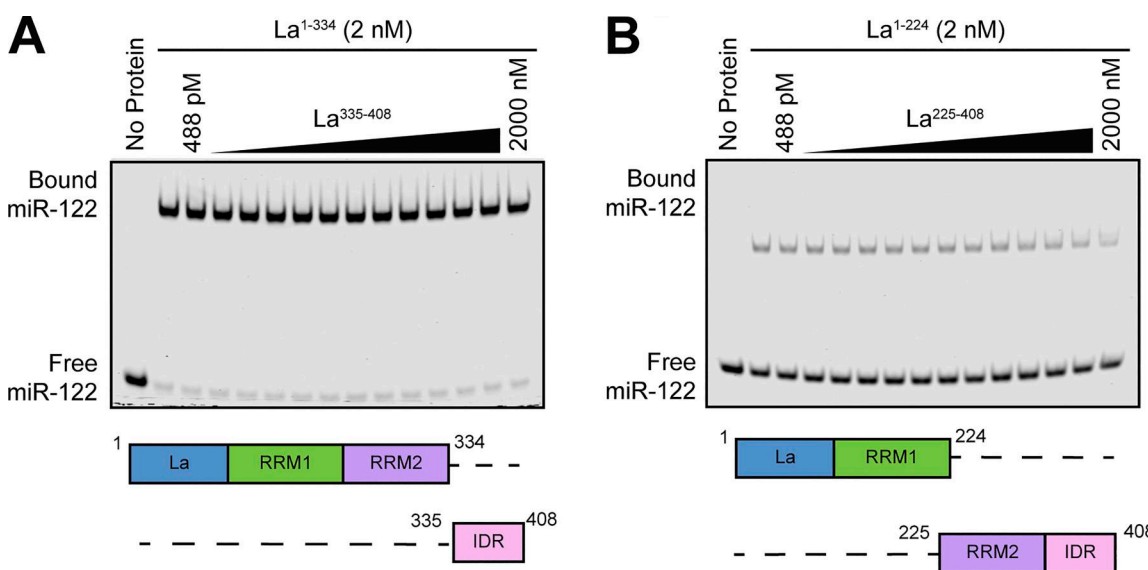

**Figure S4. Competition EMSAs between truncated La proteins. (A)** EMSA with 5' fluorescently labeled miR-122, 2 nM La$^{1-334}$, and increasing concentrations of La$^{335-408}$. La$^{335-408}$ was titrated from 448 pM to 2 μM. miR-122 migration was detected using in-gel fluorescence. **(B)** EMSA with 5' fluorescently labeled miR-122, 2 nM La$^{1-224}$, and increasing concentrations of La$^{225-408}$. La$^{225-408}$ was titrated from 448 pM to 2 μM. miR-122 migration was detected using in-gel fluorescence. Source data are available for this figure: SourceData FS4.

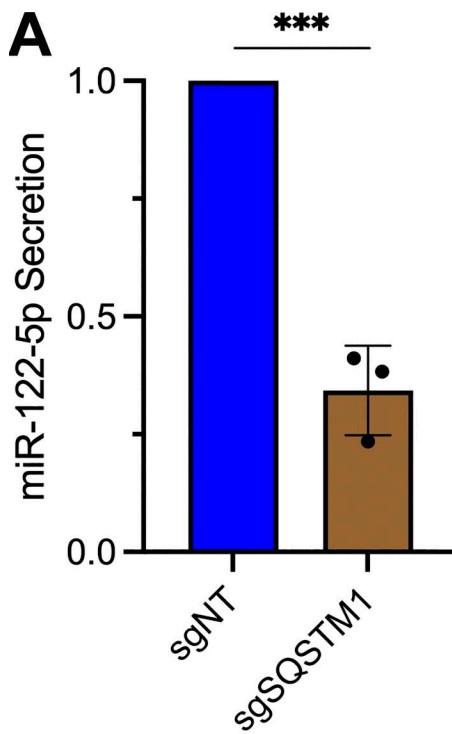

**Figure S5. p62 is required for the secretion of miR-122. (A)** RT-qPCR analysis of miR-122 from HSP fractions isolated from the conditioned medium of MDA-MB-231 cells expressing HA-CD63-mEGFP$^{ECL1}$, ZIM3-KRAB-dCas9, and either sgNT or sgSQSTM1. Statistical significance was performed using an unpaired two-tailed *t* test (*n* = 3; ***P < 0.001).

