## [Peer Review File · The Journal of Cell Biology]

p62 sorts Lupus La and selected microRNAs into breast cancer-derived exosomes

Jordan Ngo, Justin Williams, Morayma Temoche-Diaz, Abinayaa Murugupandiyam, and Randy Schekman

Corresponding Author(s): Randy Schekman, University of California, Berkeley

Review Timeline:

Submission Date:	2025-03-20
Editorial Decision:	2025-05-20
Revision Received:	2025-10-03
Editorial Decision:	2025-10-24
Revision Received:	2025-11-18

Monitoring Editor: Li Yu

Scientific Editor: Andrea Marat

Transaction Report:

DOI: <https://doi.org/10.1083/jcb.202503087>

May 20, 2025

Re: JCB manuscript #202503087

Randy Schekman
University of California, Berkeley

Dear Dr. Schekman,

Thank you for submitting your manuscript entitled "p62 sorts Lupus La and selected microRNAs into breast cancer-derived exosomes". The manuscript was assessed by expert reviewers, whose comments are appended to this letter. We invite you to submit a revision if you can address the reviewers' key concerns, as outlined here.

You will see that the reviewers are overall positive about the potential impact of your study. They have provided constructive suggestions, which we hope you agree will further improve your manuscript and that you will be able to address with experimental revisions. In particular, it is essential to completely address all comments asking for additional controls, clarifications, and quantifications. However, a detailed mechanistic extension examining a functional role in cell proliferation (reviewer 2 point 2) seems outside the scope of what is required for the current manuscript for JCB. Therefore, we hope you will be able to resolve that question in a follow up study.

GENERAL GUIDELINES:

Text limits: Character count for an Article is < 40,000, not including spaces. Count includes title page, abstract, introduction, results, discussion, and acknowledgments. Count does not include materials and methods, figure legends, references, tables, or supplemental legends.

Figures: Articles may have up to 10 main text figures. Figures must be prepared according to the policies outlined in our Instructions to Authors, under Data Presentation, <https://jcb.rupress.org/site/misc/ifora.xhtml>. All figures in accepted manuscripts will be screened prior to publication.

Supplemental information: There are strict limits on the allowable amount of supplemental data. Articles may have up to 5 supplemental figures. Up to 10 supplemental videos or flash animations are allowed. A summary of all supplemental material should appear at the end of the Materials and methods section.

Please note that JCB now requires authors to submit Source Data used to generate figures containing gels and Western blots with all revised manuscripts. This Source Data consists of fully uncropped and unprocessed images for each gel/blot displayed in the main and supplemental figures. For assays performed using capillary electrophoresis and/or immunoassay-based detection, authors should instead provide the electropherogram graph(s) for each experiment, plotting fluorescence/chemiluminescence intensity vs. molecular weight/size. Please be sure to provide one Source Data file for each figure gels, blots, and/or capillary electrophoresis assays along with your revised manuscript files. File names for Source Data figures should be alphanumeric without any spaces or special characters (i.e., SourceDataF#, where F# refers to the associated main figure number or SourceDataFS# for those associated with Supplementary figures). For traditional gels and blots, the lanes of the gels/blots should be labeled as they are in the associated figure, the place where cropping was applied should be marked (with a box), and molecular weight/size standards should be labeled wherever possible. For capillary electrophoresis assays, each trace in the graph should be color-coded and labeled to indicate which protein, gene, or sample is being measured (please try to avoid red/green combinations to accommodate our color-blind readers).

The typical timeframe for revisions is three to four months. If you anticipate any difficulties in meeting this aforementioned

revision time limit, please contact us and we can work with you to find an appropriate time frame for resubmission. Please note that papers are generally considered through only one revision cycle, so any revised manuscript will likely be either accepted or rejected.

Thank you for this interesting contribution to Journal of Cell Biology. You can contact us at the journal office with any questions at cellbio@rockefeller.edu.

Sincerely,

Li Yu, PhD
Monitoring Editor

Andrea L. Marat, PhD
Deputy Editor

Journal of Cell Biology

Reviewer #1 (Comments to the Authors (Required)):

Ngo et al. proposes a model in which p62/SQSTM1 functions as an autophagy cargo receptor for the RNA binding protein La in the LDELS pathway described by Leidal, Debnath and colleagues. The manuscript demonstrates interactions between p62 and La, both within cells and extracellularly in EV fractions. Furthermore, in both of these compartments, the authors demonstrate that p62 is partially protease protected, supporting that p62 resides in the lumen of both MVBs intracellularly as well as in purified EVs via a sedimentation and GFP-nanobody capture protocol. Although RBPs have been demonstrated to be secreted via the LDELS pathway, many of the targets do not possess an LC3-interacting motif; as a result, the employment of cargo receptors, such as p62, for LDELS provides new insight into cargo selection in this secretory autophagy pathway. This is an important finding for the field of secretory autophagy, but additional controls, clarifications or quantifications should be provided to improve the rigor of the data and the conclusions that are drawn.

- 1) For immunoblotting studies throughout the paper, it is unclear how many bio-replicates and how many have been performed and no quantification of results is provided. This is important for interpreting the differential centrifugation and the protease protection assays. Overall, Fig. 1A and B, Fig. 3A and B, Fig 4H, Fig. 5A and B, and Fig 7A-D will benefit from the quantification of multiple independent biological replicates and statistical analysis (if appropriate).
- 2) Is extracellular La in EVs subject to protease protection? This data should be included as part of Fig. 4H in addition to the current results for p62.
- 3) In Fig. 5, does LC3B or another ATG8 family member co-immunoprecipitate with La, either intracellularly and/or in EVs? This would be consistent with the proposed model in Fig. 9.
- 4) In Fig. 7D, the protein levels of LC3-II and p62 should be measured in the EV fractions (HSP) of control vs. ATG7KO cells.
- 5) In Fig. 7, although the results in ATG7KO cells are consistent with La secretion via LDELS, additional components should be functionally interrogated. LDELS has been proposed a process that is akin to CASM (conjugation of ATG8 to single membranes). ATG14 (or alternatively, FIP200 KO) cells should be analyzed to clarify whether La secretion occurs by a broader repertoire of classical autophagy components versus a CASM-like process.

Minor:

- 1) No detailed methods are provided for the generation of ATG7KO cells in Fig. 7. Is this also generated via CRISPR/Cas 9?
- 2) Is Fig 8C a single biological replicate? The methods or figure legends need to clarify the number of independent replicates analyzed.

Reviewer #2 (Comments to the Authors (Required)):

The manuscript of Ngo et al. describes studies investigating how Lupus La protein and select miRNAs are incorporated into breast cancer cell exosomes. This work builds upon prior research from the Schekman lab revealing that the RNA-binding protein La regulates the secretion of miR-122 in a subpopulation of exosomes. In this study, they employ rigorous biochemical, cell biological and profiling approaches to demonstrate that La is packaged into late endosomes and exosomes with the

autophagy cargo receptor p62 (aka SQSTM1) and that p62 is necessary for exosome secretion of La as well as multiple miRNAs including miR-122. Collectively, this work identifies p62 as an important component of the machinery that segregates cargo into exosomes and expands upon the role of the autophagy pathway in exosome secretion. While I find the study quite compelling, I believe the Ngo et al. could provide a little more granularity on the p62-La interaction as well as the functional significance of this secretory mechanism.

1. The authors should define the regions/motifs in p62 and La that required for interaction. p62 can to bind ubiquitinated cargoes to mediate their capture and degradation in the classical autophagy pathway. Ubiquitination is also implicated in cargo sorting at endosomes. However, based on the molecular weight of La in immunoblots, the pool of La packaged and secreted in exosomes does not appear to bear ubiquitin modification, suggesting ubiquitin is not involved. Is this correct? If so, what region/motifs within p62 and La mediate interaction Do p62 mutants that disrupt La interaction also result in impaired exosome secretion of La and miR-122?
2. Ngo et al speculate that p62-dependent secretion of La and select miRNAs may be a mechanism to eliminate tumor suppressor miRNAs and promote cell proliferation. This is a logical hypothesis that should be tested to examine the functional significance of p62-dependent exosome secretion in cancer cell biology. Do p62 deficient breast cancer cells have reduced proliferation compared to wild-type controls? Does rescue with wild-type p62 restore proliferation rates? What about cells rescued with p62 mutants that are defective for La binding? Perhaps there are also other ways to test this hypothesis focusing on other components of their circuit?
3. It would helpful if the authors quantified the proportion of late endosomes (CD63+) that bear luminal La and p62 across multiple cells using Airyscan micrographs. How frequently do p62 and La colocalize together?
4. In their model, they show LC3 (ATG8) also helps facilitate the packaging of p62, La and miRNAs into exosomes. Indeed, their ATG7 KO data would support this interpretation. To further support the role of LC3 in their pathway, Ngo et al should probe for LC3 in nanobody isolated CD63+ exosomes or perhaps examine LC3 localization at late endosomes via immunostaining and Airyscan super-resolution microscopy.

Reviewer #3 (Comments to the Authors (Required)):

In a previous study the authors found that the Lupus LA protein mediates sorting of miR-122 into metastatic breast cancer-derived exosomes. In this manuscript the authors study how the La protein is packaged into exosomes. Using proximity labeling proteomics, biochemical fractionation, super resolution, CRISPRi and microscopy the authors make a case that p62, a quality control factor and autophagy receptor, aka SQSTM1, modulates the miRNA composition of exosomes. Physiologically the authors propose that P62-mediated sorting of LA-bound miRNAs. While the biochemical experimentation is state of the art, the interpretation of the imaging results are not always in line with the provided data. Indeed quantitation of their images is often lacking. In addition, the physiological consequence of this sorting mechanism of miR122 is at best clear and is likely context dependent. Indeed, it remains to be determined how generalizable the observations of p62 are in other cell types, notably neuronal cells.

Main Issues

- In general, the use of the term 'exosomes' is discouraged. According to MISEV criteria the origin of the EVs should be demonstrated. It is unclear from their data that the authors are studying exosomes. The authors use MDA-MB-231 cells for Proximity labeling but aim to confirm results in HEK293T cells. While MDA cells likely produce both exosomes and ectosomes/MVs, in general HEK cells produce very few exosomes (PMID: 38718108 and PMID: 34282141) Indeed, in the introduction the authors mention that classes of microvesicles are ectosomes, apoptotic bodies and protrusion derived vesicles. However, also according to MISEV criteria, ectosomes and microvesicles are synonyms.
- To demonstrate a role for bona fide 'exosomes', pHluorin studies with P62 and or La, demonstrating actual release from internal compartments would be required. (Simply blocking exosome specific pathways is exceedingly difficult and commonly used inhibitors are highly cell-type dependent and thus not proof of sorting into exosomes.)
- The authors are referring to the buoyant density gradient and 'exosome' selection with magnetic beads, to several papers including Shurtleff et al., 2016, which states; we define exosomes as ~30-100 nm vesicles with a density of 1.08-1.18 g/ml and containing the tetraspanin protein CD63.
- As mentioned in the MISEV criteria, density and surface markers are similar between various EV subpopulations. For example, CD63 is not present on all EVs and is also expressed on ectosomes. You can only refer to exosomes when proving their endosomal origin. In this paper, with this isolation method, I recommend referring to these EVs as high density small EVs.
- In their proximity labeling assays with LA-APEX, the authors find P62 as binding partner. But they also find Hur and hnRNPA2B1, these are known to bind selected miRNAs can the authors demonstrate enrichment of these miRNAs (miR17 and or others) showing the effect is microRNA specific?
- The sorting of La in their sEVs is convincing, yet it may be cell-type specific for example La appears absent from exosomes from LCL cells (PMID: 26768848). The authors should reconcile or mention that their observations may be context dependent.

- Stoichiometry; It has been estimated that human cells express $\sim 2 \times 10^7$ copies of the La protein. The La protein dominant function is binding to tRNAs (3'UUU) that are most likely much more abundant in the cytoplasm than miR122. How many copies of miR122 are present per cell and per sEV? Can this be reconciled with the copies of La protein per EV?
- To prove that part of the La and P62 is present in exosomes, I would like to see immunogold labelling in an MVB. This makes exosomal secretion highly likely (but does not prove that La+ MVBs are secreted, as some MVBs are also degraded). The IF images on p62 and La in late endosomes alone are not convincing enough that the p62 and La in EVs is (at least partly) coming from the endolysosomal route
- In many microscopy images, the authors show just 1 cell. To increase reliability, it is recommended to include either a bigger FOV and then zoom in on 1 cell, or include additional images of single cells in the supplementary.
- In their comparison study, when sequencing MDA-MB-231 cells and exosomes, the authors show that P62KO cells and exosomes are different from wt in miRNA composition (Fig 8). Was the normalization done by counting EVs? The authors choose not to overexpress p62 or induce its expression for example by Bafilomycin in non- low-p62 expressing cells (PMID: 35446347). Would one still observe this increase in sorting?
- The authors acknowledge that P62 is also secreted in non-EV form, is La also secreted in not-bound to EVs?
- The schematic should be adjusted to the outcome of the additional experiments and more nuanced, not every sEV will contain this complex, nor is it clear (yet) that the sorting has place at endosome level.

- Figure 1

- Why is there a white line in the smear of CD63 on the WB of fig 1A
- I'm wondering how representative the authors think the image of figure 1D is of the general population. Based on my own CD63 stainings in MDA cells, I know that such a large late endosome as shown in this image is present in about 5-10% of the cells. I would be interested in how the authors explain this, as also in this image we don't see any LA accumulation occurring in the smaller CD63 compartments in this cell.

Figure 3

- The background signal of La in Figure 3C is higher (with even nuclear staining) compared to the La staining of Fig 1C&D and Fig S1A. I would recommend choosing a different cell
- It is phrased correctly that a portion of p62 is captured together with La into the lumen of CD63 positive late endosomes, but the proof is scarce. We see one p62 puncta in one cell in a CD63 late endosome, while the remaining CD63+ late endosomes are devoid of p62.

Figure 4

- eGFP-CD63 capture does not mean you capture exosomes, other EV subtypes MVs and ectosomes are also CD63 positive with a similar topology (and can have a similar density).

Dear Dr. Yu and Dr. Marat,

Thank you for your consideration of our manuscript for publication in the *Journal of Cell Biology* and for providing us the opportunity to address the comments and suggestions of the reviewers. We also thank the three expert reviewers for their constructive and thoughtful suggestions. Their feedback led us to perform additional experiments which have strengthened our manuscript.

Here is a summary of the main revisions included in our updated manuscript:

1. We tested whether knockout of FIP200 blocks the secretion of La, p62 and LC3.
2. We tested whether knockout of ATG7 affects the secretion of p62 and LC3.
3. We evaluated whether LC3 is present in cellular La immunoprecipitates.
4. We conducted *in vitro* binding assays to test whether La interacts directly with p62.
5. We conducted Airyscan immunofluorescence experiments to evaluate the presence of LC3 in CD63-positive late endosomes.
6. We quantified the immunoblotting data of our membrane fractionation, protease protection, and EV secretion experiments.
7. We modified the text to include additional clarifications and caveats.

We have included our point-by-point response to the three expert reviewers below. Thank you again for your consideration of our manuscript.

Reviewer #1 (Comments to the Authors (Required)):

Ngo et al. proposes a model in which p62/SQSTM1 functions as an autophagy cargo receptor for the RNA binding protein La in the LDELS pathway described by Leidal, Debnath and colleagues. The manuscript demonstrates interactions between p62 and La, both within cells and extracellularly in EV fractions. Furthermore, in both of these compartments, the authors demonstrate that p62 is partially protease protected, supporting that p62 resides in the lumen of both MVBs intracellularly as well as in purified EVs via a sedimentation and GFP-nanobody capture protocol. Although RBPs have been demonstrated to be secreted via the LDELS pathway, many of the targets do not possess an LC3-interacting motif; as a result, the employment of cargo receptors, such as p62, for LDELS provides new insight into cargo selection in this secretory autophagy pathway. This is an important finding for the field of secretory autophagy, but additional controls, clarifications or quantifications should be provided to improve the rigor of the data and the conclusions that are drawn.

1) For immunoblotting studies throughout the paper, it is unclear how many bio-replicates and have been performed and no quantification of results is provided. This is important for interpreting the differential centrifugation and the protease protection assays. Overall, Fig. 1A and B, Fig. 3A and B, Fig 4H, Fig. 5A and B, and Fig 7A-D will benefit from the quantification of multiple independent biological replicates and statistical analysis (if appropriate).

We thank the reviewer for this important point. We have clarified the number of biological replicates in the figure legends and have added additional panels quantifying our immunoblotting data (new Figures 1B, 1D, 3B, 3D, 4I, 7C, 7F, 7I).

2) Is extracellular La in EVs subject to protease protection? This data should be included as part of Fig. 4H in addition to the current results for p62.

Great question. Yes, we previously demonstrated that extracellular La is resistant to proteolysis in the absence of detergent (Temoche-Diaz et al., 2019; Author Response Image 1). We have added an additional sentence to the results section to highlight our previous data.

Author Response Image 1. Immunoblot analysis of proteinase K protection experiments conducted on HSP fractions from MDA-MB-231 cells (From Temoche-Diaz et al., 2019; Figure 6F).

3) In Fig. 5, does LC3B or another ATG8 family member co-immunoprecipitate with La, either intracellularly and/or in EVs? This would be consistent with the proposed model in Fig. 9.

Good question. We evaluated whether LC3B was present in our cellular La immunoprecipitates and found that LC3B does co-immunoprecipitate with La. We have included these data in the revised manuscript (updated Figure 5A).

4) In Fig. 7D, the proteins levels of LC3-II and p62 should be measured in the EV fractions (HSP) of control vs. ATG7KO cells.

Good suggestion. We evaluated the secretion of LC3 and p62 in HSP fractions isolated from wild-type and ATG7 KO cells. We found that ATG7 KO blocks LC3 and p62 secretion in HSP fractions, and we have included these data in the revised manuscript (updated Figure 7E and 7F).

5) In Fig. 7, although the results in ATG7KO cells are consistent with La secretion via LDELS, additional components should be functionally interrogated. LDELS has been proposed a process that is akin to CASM (conjugation of ATG8 to single membranes). ATG14 (or alternatively, FIP200 KO) cells should be analyzed to clarify whether La secretion occurs by a broader repertoire of classical autophagy components versus a CASM-like process.

We thank the reviewer for this great suggestion. To clarify whether La secretion occurs via a broader repertoire of autophagy pathways, we obtained FIP200 knockout (KO) cells. We then isolated HSP fractions from wild-type and FIP200 KO cells for immunoblot analysis and found that FIP200 KO did not affect the secretion of La, p62 or LC3. These data provide additional evidence suggesting that La is secreted via the LDELS pathway. These data have been included in the revised manuscript (new Figure 7G-7I).

Minor:

1) No detailed methods are provided for the generation of ATG7KO cells in Fig. 7. Is this also generated via CRISPR/Cas 9?

The ATG7 knockout cells were generated via CRISPR/Cas9 in the original LDELS manuscript (Leidal et al., 2020) and were a kind gift from Dr. Jayanta Debnath (UC San Francisco). We previously indicated this in our acknowledgements section. To increase clarity, we have added an additional acknowledgement (under 'Cell lines, media, and general chemicals') stating that the ATG7 knockout cells were a kind gift from the Debnath team.

2) Is Fig 8C a single biological replicate? The methods or figure legends need to clarify the number of independent replicates analyzed.

The data in Figure 8C were obtained from three biological replicates. We have updated the text in both the methods section and figure legend to clarify this.

Reviewer #2 (Comments to the Authors (Required)):

The manuscript of Ngo et al. describes studies investigating how Lupus La protein and select miRNAs are incorporated into breast cancer cell exosomes. This work builds upon prior research from the Schekman lab revealing that the RNA-binding protein La regulates the secretion of miR-122 in a subpopulation of exosomes. In this study, they employ rigorous biochemical, cell biological and profiling approaches to demonstrate that La is packaged into late endosomes and exosomes with the autophagy cargo receptor p62 (aka SQSTM1) and that p62 is necessary for exosome secretion of La as well as multiple miRNAs including miR-122. Collectively, this work identifies p62 as an important component of the machinery that segregates cargo into exosomes and expands upon the role of the autophagy pathway in exosome secretion. While I find the study quite compelling, I believe the Ngo et al. could provide a little more granularity on the p62-La interaction as well as the functional significance of this secretory mechanism.

1. The authors should define the regions/motifs in p62 and La that required for interaction. p62 can to bind ubiquitinated cargoes to mediate their capture and degradation in the classical autophagy pathway. Ubiquitination is also implicated in cargo sorting at endosomes. However, based on the molecular weight of La in immunoblots, the pool of La packaged and secreted in exosomes does not appear to bear ubiquitin modification, suggesting ubiquitin is not involved. Is this correct? If so, what region/motifs within p62 and La mediate interaction Do p62 mutants that disrupt La interaction also result in impaired exosome secretion of La and miR-122?

We thank the reviewer for this thoughtful point. We were also interested in this question given the molecular weight of La in exosomes. To test whether p62 interacts directly with La, we purified His₆-MBP-p62 from *E. coli* for use in FLAG IP experiments with recombinant La-Nluc-FLAG-His₆. In this binding reaction, we added significantly lower concentrations of La and p62 compared to their cellular concentrations as described in the OpenCell database (Cho et al., 2022). Nevertheless, La and p62 still bound to each other, indicating that the K_d of the La:p62 interaction is lower than the physiological concentration of these proteins. We thus conclude that La binds directly to p62. We have added these data to the revised manuscript (new Figure 5C), and we have updated the discussion section to include these results.

We are also very interested in identifying the binding interface between La and p62 and evaluating whether disrupting this interaction interface affects exosomal miRNA secretion. We would like to reserve these experiments for a follow-up study focused on elucidating the role p62-dependent exosomal secretion on tumor cell proliferation.

2. Ngo et al speculate that p62-dependent secretion of La and select miRNAs may be a mechanism to eliminate tumor suppressor miRNAs and promote cell proliferation. This is a logical hypothesis that should be tested to examine the functional significance of p62-dependent exosome secretion in cancer cell biology. Do p62 deficient breast cancer cells have reduced proliferation compared to wild-type controls? Does rescue with wild-type p62 restore proliferation rates? What about cells rescued with p62 mutants that are defective for La binding? Perhaps there are also other ways to test this hypothesis focusing on other components of their circuit?

We thank the reviewer for these fantastic suggestions. We are also keenly interested in assessing the effect of p62-dependent exosomal miRNA secretion on tumor cell proliferation. We would like to reserve these experiments for a follow-up study focused on this important question.

3. It would be helpful if the authors quantified the proportion of late endosomes (CD63+) that bear luminal La and p62 across multiple cells using Airyscan micrographs. How frequently do p62 and La colocalize together?

Great suggestion. Because many CD63-positive late endosomes were too small to evaluate the presence of luminal La and p62 (even while using Airyscan super-resolution microscopy), we have quantified the immunoblotting data from our membrane fractionation and protease protection experiments to assess the amount of La and p62 present within membrane-bound compartments (new Figure 1B, 1D, 3B, and 3D). From these quantifications, it appears that ~4% of La contained within the 20k membrane fraction (corresponding to ~2.5% of cytoplasmic, sedimentable La and <0.1% of total La) is captured into the lumen of a membrane-bound organelle. We have edited the text of the results section to highlight these quantifications. We hope that these quantifications satisfy the reviewer.

4. In their model, they show LC3 (ATG8) also helps facilitate the packaging of p62, La and miRNAs into exosomes. Indeed, their ATG7 KO data would support this interpretation. To further support the role of LC3 in their pathway, Ngo et al should probe for LC3 in nanobody isolated CD63+ exosomes or perhaps examine LC3 localization at late endosomes via immunostaining and Airyscan super-resolution microscopy.

Good suggestion. We have conducted additional immunofluorescence experiments to visualize the co-localization of LC3 at CD63-positive late endosomes via immunostaining and Airyscan super-resolution microscopy. We visualized LC3B and p62 dual-positive puncta at both the limiting membrane and within the lumen of CD63-positive late endosomes (new Figure 3G and 3H). We have added these data to the revised manuscript.

Reviewer #3 (Comments to the Authors (Required)):

In a previous study the authors found that the Lupus LA protein mediates sorting of miR-122 into metastatic breast cancer-derived exosomes. In this manuscript the authors study how the La protein is packaged into exosomes.

Using proximity labeling proteomics, biochemical fractionation, super resolution, CRISPRi and microscopy the authors make a case that p62, a quality control factor and autophagy receptor, aka SQSTM1, modulates the miRNA composition of exosomes. Physiologically the authors propose that P62-mediated sorting of LA-bound miRNAs. While the biochemical experimentation is state of the art, the interpretation of the imaging results are not always in line with the provided data. Indeed quantitation of their images is often lacking. In addition, the physiological consequence of this sorting mechanism of miR122 is at best unclear and is likely context dependent. Indeed, it remains to be determined how generalizable the observations of p62 are in other cell types, notably neuronal cells.

Main Issues

- In general, the use of the term 'exosomes' is discouraged. According to MISEV criteria the origin of the EVs should be demonstrated. It is unclear from their data that the authors are studying exosomes.

We appreciate the reviewer's consideration of EV nomenclature. We believe that our data, published both in Temoche-Diaz et al., 2019 and described here in this manuscript, provide strong evidence that we are indeed monitoring the secretion of late endosome-derived EVs (exosomes).

Previously, in Temoche-Diaz et al., 2019, we employed buoyant density gradient flotation through a high-resolution iodixanol gradient and found that, at least in MDA-MB-231 cells, a majority of CD63 fractionates away from a low-buoyant density EV subpopulation that is characterized by enrichment of CD9 and other plasma membrane proteins such as MFGE-8 (Temoche-Diaz et al., 2019; Author Response Image 2).

Author Response Image 2. Immunoblot analysis of EV markers after flotation of EVs through a high-resolution, linear iodixanol gradient (From Temoche-Diaz et al., 2019; Figure 1B)

Additionally, we demonstrated that La partitions to these high buoyant density EVs (Temoche-Diaz et al., 2019; Author Response Image 3).

Author Response Image 3. Immunoblot analysis of EV markers, La, and NCL after flotation through a high-resolution, linear iodixanol gradient (From Temoche-Diaz et al., 2019; Figure 6 – figure supplement 1).

We also showed that the secretion of this high buoyant density population of CD63-positive EVs was specifically depleted upon knockout of Rab27a, a Rab GTPase that is well characterized to be required for exosome secretion (Ostrowski et al., 2010), but not knockout of Rab35 (Temoche-Diaz et al., 2019; Author Response Image 4). Importantly, knockout of Rab27a had no effect on CD9 secretion. Our genetic knockout experiments are consistent with a previous study published by the labs of Clotilde Théry and Graca Raposo (Bobbie et al., 2012). Overall, our results demonstrate that, at least in MDA-MB-231 cells, La is secreted within exosomes.

Author Response Image 4. (Left) Immunoblot analysis confirming successful knockout of Rab27a and Rab35 in MDA-MB-231 cells. (Right) Immunoblot analysis demonstrating that knockout of Rab27a blocks the secretion of CD63-positive EVs, but not CD9-positive EVs, from MDA-MB-231 cells. (From Temoche-Diaz et al., 2019; Figure 2E and 2F).

In the current manuscript, we sought to follow up on our previous data and elucidate the mechanism by which La is secreted. We developed a CD63 construct that is similar to the well-established CD63-pHluorin fusion protein (Verweij et al., 2018) that has been well demonstrated to localize to late endosomes/MVBs. In further support of this proper localization, we were unable to detect GFP signal from these cells under basal growth conditions. We then deacidified lysosomes via treatment with Bafilomycin A1 (BafA1) and saw a dramatic increase in GFP fluorescence, demonstrating that our HA-CD63-mEGFP^{ECL1} construct correctly traffics to and resides within acidified, late endosomal compartments. Furthermore, immunoblot analysis of our GFP nanobody extracellular immunoprecipitates indicated distinct enrichment of both CD63 and p62, and considerable dis-enrichment of ANXA2, an established microvesicle marker (Jeppesen et al., 2019; Williams et al., 2025). Additionally, we have provided genetic evidence that the secretion of La and p62 occurs in an ATG7-dependent and FIP200-independent manner. This is consistent with the LC3-dependent EV loading and secretion (LDELS) pathway, which occurs via a CASM-like process.

On the basis of these experimental results, we believe that we are studying the secretion of exosomes and that our data are consistent with the conclusions determined by Mathieu et al., 2022 (PMID: 34282141). Indeed, Mathieu et al., 2022 directly state at the beginning of their discussion section: “In this work, we provide evidence that sEVs bearing tetraspanins, especially CD9 and CD81 with little CD63, bud mainly from the plasma membrane, whereas others bearing CD63 with little CD9 but containing some late endosome proteins form in internal compartments and qualify as exosomes.”

The authors use MDA-MB-231 cells for Proximity labeling but aim to confirm results in HEK293T cells.

We found that p62 depletion in MDA-MB-231 cells impairs the secretion of La within GFP nanobody-isolated CD63-positive exosomes (Figure 7A-7C).

While MDA cells likely produce both exosomes and ectosomes/MVs, In general HEK cells produce very few exosomes (PMID: 38718108 and PMID: 34282141) Indeed, in the introduction the authors mentions that classes of microvesicles are ectosomes, apoptotic bodies and protrusion derived vesicles. However, also according to MISEV criteria, ectosomes and microvesicles are synonyms.

We apologize for the textual error regarding microvesicles and ectosomes. We have corrected this in the introduction section of the revised manuscript.

- To demonstrate a role for bona fide 'exosomes', pHluorin studies with P62 and or La, demonstrating actual release from internal compartments would be required. (Simply blocking exosomes specific pathways is

exceedingly difficult and commonly used inhibitors are highly cell-type dependent and thus not proof of sorting into exosomes.)

Good suggestion. However, we cannot conduct these pHluorin experiments with La or p62. pHluorin exocytosis studies are based upon the 'flash' that occurs upon rapid pH neutralization (e.g. when an acidic compartment fuses with the plasma membrane). Thus, these pHluorin studies are primarily amenable for transmembrane proteins that access the lumen of an acidified compartment (such as with CD63). La and p62 do not primarily localize to acidic compartments, and thus any signal derived from pHluorin flashes will be masked by a large background signal derived from the primary populations of La and p62 that are present at neutral pH.

- The authors are referring to the buoyant density gradient and 'exosome' selection with magnetic beads, to several papers including Shurtleff et al., 2016, which states; we define exosomes as ~30-100 nm vesicles with a density of 1.08-1.18 g/ml and containing the tetraspanin protein CD63.

While we cite Shurtleff et al., 2016 in our paper, we also state in our introduction section that the goal of this manuscript was to follow up on our previous data from Temoche-Diaz et al., 2019 and identify the molecular mechanism by which La is secreted. These experiments were based upon our previous result that extracellular La fractionates to a high buoyant density EV subpopulation that is enriched with CD63 and disenriched of CD9 (described in further detail in our response to the first point of this reviewer).

- As mentioned in the MISEV criteria, density and surface markers are similar between various EV subpopulations. For example, CD63 is not present on all EVs and is also expressed on ectosomes. You can only refer to exosomes when proving their endosomal origin. In this paper, with this isolation method, I recommend referring to these EVs as high density small EVs.

Our response to this point is included in our response to the first point of this reviewer.

- In their proximity labeling assays with LA-APEX, the authors find P62 as binding partner. But they also find Hur and hnRNPA2B1, these are known to bind selected miRNAs can the authors demonstrate enrichment of these miRNAs (miR17 and or others) showing the effect is microRNA specific?

Good point. We have analyzed our small RNA sequencing datasets to evaluate the effect of p62 depletion on the intracellular accumulation and exosomal secretion of miR-17. We observed that p62 depletion led to a slight increase in miR-17 both in cells and in exosomes, contrary to what we observed for miR-122 and other p62-dependent exosomal miRNAs (an increase in cells and decrease in exosomes). We have included these data in the revised manuscript (updated Figure 8C) and updated the text of the results section to include miR-17.

- The sorting of La in their sEVs is convincing, yet it may be cell-type specific for example La appears absent from exosomes from LCL cells (PMID: 26768848). The authors should reconcile or mention that their observations may be context dependent.

Good point. We have added an additional statement in the discussion section to clarify that our results may be context-dependent.

- Stoichiometry; It has been estimated that human cells express $\sim 2 \times 10^7$ copies of the La protein. The La protein dominant function is binding to tRNAs (3'UUU) that are most likely much more abundant in the cytoplasm than miR122. How many copies of miR122 are present per cell and per sEV? Can this be reconciled with the copies of La protein per EV?

We appreciate the reviewer's consideration of cytoplasmic tRNA stoichiometry. However, 3'UUU removal and subsequent 3'CCA addition are required for the export of pre-tRNAs from the nucleus to the cytoplasm

(Jouravleva and Zamore, 2025). As such, one would not expect that 3'UUU-bearing tRNAs to be present at high levels in the cytoplasm where they could potentially compete with miR-122 for cytoplasmic La binding. Indeed, 3'UUU removal from nuclear pre-tRNAs prior to nuclear export actually confers upon cytoplasmic La the ability to bind other polyuridine-containing RNAs, as has been previously described for the ability of La to interact with other RNA molecules such as 5' terminal oligopyrimidine (5'TOP) RNAs (Crosio, 2000; Cardinali et al., 2003; Intine et al., 2003).

- To prove that part of the La and P62 is present in exosomes, I would like to see immunogold labelling in an MVB. This makes exosomal secretion highly likely (but does not prove that La+ MVBs are secreted, as some MVBs are also degraded). The IF images on p62 and La in late endosomes alone are not convincing enough that the p62 and La in EVs is (at least partly) coming from the endolysosomal route

Our response to this point is included in our response to the first point of this reviewer.

- In many microscopy images, the authors show just 1 cell. To increase reliability, its recommended to include either a bigger FOV and then zoom in on 1 cell, or include additional images of single cells in the supplementary.

Good point. We had to zoom into single cells to achieve the resolution required to visualize the co-localization of two proteins within the lumen of large, CD63-positive late endosomes. To address this point, we have included additional images in the supplementary (new Figure S1).

- In their comparison study, when sequencing MDA-MB-231 cells and exosomes, the authors show that P62KO cells and exosomes are different from wt in miRNA composition (Fig 8). Was the normalization done by counting EVs? The authors choose not to overexpress p62 or induce its expression for example by Bafilomycin in non- low-p62 expressing cells (PMID: 35446347). Would one still observe this increase in sorting?

Good question. The small RNA library preparation was normalized using the same amount of input RNA as quantified using an Agilent 2100 Bioanalyzer. The sequencing reads were mapped using miRDeep2, and the mapped miRNA reads were normalized by dividing the reads per miRNA to the total number of miRNA reads per sample. This is a standard method of normalization for RNA sequencing experiments.

We chose not to overexpress p62 to avoid any potential overexpression artifacts. We also chose to not monitor EV secretion upon BafA1 treatment because this results in the secretion of a primarily protease-accessible form of p62 (Solvik et al., 2022) which is derived from amphisome exocytosis.

- The authors acknowledge that P62 is also secreted in non-EV form, is La also secreted in not-bound to EVs?

Good question. We demonstrated in our previous manuscript that extracellular La is resistant to proteolysis in the absence of detergent (Temoche-Diaz et al., 2019; Author Response Image 1). We have modified the text to highlight our previous result.

- The schematic should adjusted to the outcome of the addiioatnal experiments and more nuanced, not every sEV will contain this complex, nor is it clear (yet) that the sorting has place at endosome level.

We have clarified in Figure 9 that the depicted schematic specifically illustrates p62-dependent sorting of cargos into ILVs.

Figure 1

- Why is there a white line in the smear of CD63 on the WB of fig 1A

The differential CD63 banding pattern corresponds to different glycosylated CD63 molecules. The banding pattern appears as a smear at longer exposures.

- I'm am wondering how representable the authors think the image of figure 1D is of the general population. Based on my own CD63 stainings in MDA cells, I know that such a large late endosomes as shown in this image is present in about 5-10% of the cells. I would be interesting in how the authors explain this, as also in this image we don't see any LA accumulation occurring in the smaller CD63 compartments in this cell.

Good question. We suspect that heterogeneity in late endosome size may arise due to variability in the nutrients available in conditioned medium preparations (such as different sources/preparations of FBS). From our CD63 immunostainings, we have observed that more than ~40% of cells contain these larger CD63-positive late endosomes.

Regarding the presence of La within smaller endosomes, we were unable to determine whether they contain La because the endosomes were too small to analyze their luminal content via our immunofluorescence experiments. Additionally, we have quantified our membrane fractionation and protease protection experiments (New Figures 1B, 1D, 3B, and 3D) to determine the amount of membrane protected La and p62. From these quantifications, it appears that ~4% of La contained within the 20k membrane fraction (corresponding to ~2.5% of cytoplasmic, sedimentable La and <0.1% of total La) is captured into the lumen of a membrane-bound organelle. Thus, we would expect the presence of La within a membrane-bound compartment to be a rare event. We have updated the text of the results section to include our quantifications and clarify these points.

Figure3

- The background signal of La in Figure 3C is higher (with even nuclear staining) compared to the La staining of Fig 1C&D and Fig S1A. I would recommend choosing a different cell

Despite the increased background signal, we were still able to visualize cytoplasmic La puncta in these imaging experiments.

- It is phrased correctly that a portion of p62 is captured together with La into the lumen of CD63 positive late endosomes, but the prove is scarce. We see one p62 puncta in one cell in a CD63 late endosomes, while the remaining CD63+ late endosomes are devoid of p62.

We have quantified our membrane fractionation and protease protection experiments in our revised manuscript, and from these data, we would not expect the co-localization of La and p62 puncta within CD63-positive late endosomes to be a common event (given that only ~4% of cytoplasmic sedimentable La is captured within a membrane-protected compartment). We have modified the text of the results section to include the quantification of our membrane fractionation and protease protection experiments and clarify that we were observing a rare event.

Figure 4

- eGFP-CD63 capture does not mean you capture exosomes, other EV subtypes MVs and ectosomes are also CD63 positive with a similar topology (and can have a similar density).

Our response to this point is included in our response to the first point of this reviewer.

October 24, 2025

RE: JCB Manuscript #202503087R

Randy Schekman
University of California, Berkeley

Dear Dr. Schekman:

Thank you for submitting your revised manuscript entitled "p62 sorts Lupus La and selected microRNAs into breast cancer-derived exosomes". We would be happy to publish your paper in JCB pending final revisions necessary to meet our formatting guidelines (see details below).

A. MANUSCRIPT ORGANIZATION AND FORMATTING:

- 1) Text limits: Character count for Articles is < 40,000, not including spaces. Count includes abstract, introduction, results, discussion, and acknowledgments. Count does not include title page, figure legends, materials and methods, references, tables, or supplemental legends.
- 2) Figures limits: Articles may have up to 10 main text figures.
- 3) Figure formatting: * Scale bars must be present on all microscopy images, including inset magnifications (* you may alternatively indicate the diameter of insets) *. Molecular weight or nucleic acid size markers must be included on all gel electrophoresis. Aspect ratios of images may not be altered.
- 4) Statistical analysis: Error bars on graphic representations of numerical data must be clearly described in the figure legend. The number of independent data points (n) represented in a graph must be indicated in the legend. Statistical methods should be explained in full in the materials and methods. For figures presenting pooled data the statistical measure should be defined in the figure legends. Please also be sure to indicate the statistical tests used in each of your experiments (either in the figure legend itself or in a separate methods section) as well as the parameters of the test (for example, if you ran a t-test, please indicate if it was one- or two-sided, etc.). Also, if you used parametric tests, please indicate if the data distribution was tested for normality (and if so, how). If not, you must state something to the effect that "Data distribution was assumed to be normal but this was not formally tested."
- 5) Abstract and title: The abstract should be no longer than 160 words and should communicate the significance of the paper for a general audience. The title should be less than 100 characters including spaces. Make the title concise but accessible to a general readership.
- 6) Materials and methods: Should be comprehensive and not simply reference a previous publication for details on how an experiment was performed. Please provide full descriptions in the text for readers who may not have access to referenced manuscripts.
- 7) All antibodies, cell lines, animals, and tools used in the manuscript should be described in full, including accession numbers for materials available in a public repository such as the Resource Identification Portal. Please be sure to provide the sequences for all of your primers/oligos and RNAi constructs in the materials and methods. You must also indicate in the methods the source, species, and catalog numbers (where appropriate) for all of your antibodies. Please also indicate the acquisition and quantification methods for immunoblotting/western blots.
- 8) Microscope image acquisition: The following information must be provided about the acquisition and processing of images:
 - a. Make and model of microscope
 - b. Type, magnification, and numerical aperture of the objective lenses
 - c. Temperature
 - d. Imaging medium
 - e. Fluorochromes
 - f. Camera make and model
 - g. Acquisition software
 - h. Any software used for image processing subsequent to data acquisition. Please include details and types of operations

involved (e.g., type of deconvolution, 3D reconstitutions, surface or volume rendering, gamma adjustments, etc.).

10) Supplemental materials: There are strict limits on the allowable amount of supplemental data. Articles may have up to 5 supplemental figures. Please also note that tables, like figures, should be provided as individual, editable files. A summary of all supplemental material should appear at the end of the Materials and methods section.

13) ORCID IDs: ORCID IDs are unique identifiers allowing researchers to create a record of their various scholarly contributions in a single place. Please note that ORCID IDs are now *required* for all authors. At resubmission of your final files, please be sure to provide your ORCID ID and those of all co-authors.

Please note that JCB now requires authors to submit Source Data used to generate figures containing gels and Western blots with all revised manuscripts. This Source Data consists of fully uncropped and unprocessed images for each gel/blot displayed in the main and supplemental figures. For assays performed using capillary electrophoresis and/or immunoassay-based detection, authors should instead provide the electropherogram graph(s) for each experiment, plotting fluorescence/chemiluminescence intensity vs. molecular weight/size. Please be sure to provide one Source Data file for each figure gels, blots, and/or capillary electrophoresis assays along with your revised manuscript files. File names for Source Data figures should be alphanumeric without any spaces or special characters (i.e., SourceDataF#, where F# refers to the associated main figure number or SourceDataFS# for those associated with Supplementary figures). For traditional gels and blots, the lanes of the gels/blots should be labeled as they are in the associated figure, the place where cropping was applied should be marked (with a box), and molecular weight/size standards should be labeled wherever possible. For capillary electrophoresis assays, each trace in the graph should be color-coded and labeled to indicate which protein, gene, or sample is being measured (please try to avoid red/green combinations to accommodate our color-blind readers).

Journal of Cell Biology now requires a data availability statement for all research article submissions. These statements will be published in the article directly above the Acknowledgments. The statement should address all data underlying the research presented in the manuscript. Please visit the JCB instructions for authors for guidelines and examples of statements at (<https://rupress.org/jcb/pages/editorial-policies#data-availability-statement>).

B. FINAL FILES:

****It is JCB policy that if requested, original data images must be made available to the editors. Failure to provide original images upon request will result in unavoidable delays in publication. Please ensure that you have access to all original data images prior to final submission.****

****The license to publish form must be signed before your manuscript can be sent to production. A link to the electronic license to publish form will be sent to the corresponding author only. Please take a moment to check your funder requirements before choosing the appropriate license.****

Thank you for your attention to these final processing requirements. Please revise and format the manuscript and upload materials within 7 days. If you need an extension for whatever reason, please let us know and we can work with you to determine a suitable revision period.

Thank you for this interesting contribution, we look forward to publishing your paper in Journal of Cell Biology.

Sincerely,

Li Yu
Monitoring Editor

Andrea L. Marat
Deputy Editor

Journal of Cell Biology

Reviewer #1 (Comments to the Authors (Required)):

I have reviewed the revised manuscript and the authors' response. My previous concerns have been completely addressed.

Reviewer #2 (Comments to the Authors (Required)):

The revised manuscript of Ngo et al. describes studies that investigate the mechanisms that Lupus La protein and select miRNAs are incorporated into breast cancer cell exosomes. This work builds upon prior research from the Schekman lab revealing that the RNA-binding protein La regulates the secretion of miR-122 in a subpopulation of exosomes. In this study, they employ rigorous biochemical, cell biological and profiling approaches to demonstrate:

- 1) La is captured into the lumen of late endosomes (strongly supportive)
- 2) Identify p62 (aka SQSTM1) as candidate factor mediating La packaging (strongly supportive)
- 3) Demonstrate p62 is packaged with La at late endosomes and secreted in with this RBP in a subpopulation of exosomes (strongly supportive)
- 4) La bind miRNAs (strongly supportive)
- 5) p62 and autophagy pathway components are necessary for La and miR-122 secretion in exosomes (strongly supportive)

Collectively, the original and newly incorporated data provide strong evidence for cargo receptor and ATG-dependent packaging of La and miRNAs into exosomes and expands upon the role of the autophagy pathway in exosome secretion.